# MULTIPLAYER NASH PREFERENCE OPTIMIZATION

**Fang Wu**$^{\heartsuit,*}$, **Xu Huang**$^{\spadesuit,*}$, **Weihao Xuan**$^{\triangledown,\triangle}$, **Zhiwei Zhang**$^{\clubsuit}$, **Yijia Xiao**$^{\diamond}$
**Guancheng Wan**$^{\diamond}$, **Xiaomin Li**$^{\square}$, **Bing Hu**$^{\circ}$, **Peng Xia**$^{\star}$, **Jure Leskovec**$^{\heartsuit}$, **Yejin Choi**$^{\heartsuit\dagger}$
$^{\heartsuit}$Stanford University, $^{\spadesuit}$Georgia Institute of Technology, $^{\triangledown}$The University of Tokyo
$^{\triangle}$RIKEN AIP, $^{\clubsuit}$Pennsylvania State University, $^{\diamond}$University of California, Los Angeles,
$^{\square}$Harvard University, $^{\circ}$Independent Researcher, $^{\star}$UNC–Chapel Hill
`fangwu@stanford.edu`, `xu.huang@gatech.edu`, `yejin@cs.stanford.edu`

## ABSTRACT

Reinforcement learning from human feedback (RLHF) has emerged as the standard paradigm for aligning large language models with human preferences. However, reward-based methods grounded in the Bradley—Terry assumption struggle to capture the non-transitive and heterogeneous nature of real-world preferences. To address this, recent studies have reframed alignment as a two-player Nash game, giving rise to Nash learning from human feedback (NLHF). While this perspective has inspired algorithms such as INPO, ONPO, and EGPO that offer strong theoretical and empirical guarantees, they remain fundamentally restricted to two-player interactions, introducing a single-opponent bias that fails to capture the full complexity of realistic preference structures. This work introduces Multiplayer Nash Preference Optimization (MNPO), a novel framework that generalizes NLHF to the multiplayer regime. It formulates alignment as an $n$-player game, where each policy competes against a population of opponents while being regularized toward a reference model. We demonstrate that MNPO inherits the equilibrium guarantees of two-player methods while enabling richer competitive dynamics and improved coverage of diverse preference structures. Comprehensive empirical evaluation shows that MNPO consistently outperforms existing NLHF baselines on instruction-following benchmarks, achieving superior alignment quality under heterogeneous annotator conditions and mixed-policy evaluation scenarios. Together, these results establish MNPO as a principled and scalable framework for aligning LLMs with complex, non-transitive human preferences. Code is available at `https://github.com/smiles724/MNPO`.

## 1 INTRODUCTION

Large language models (LLMs) have achieved remarkable progress in instruction following and open-ended reasoning through reinforcement learning from human feedback (RLHF) (Christiano et al., 2017). Traditional RLHF pipelines built upon the *Bradley–Terry* (Bradley & Terry, 1952) model have enabled widely deployed systems (e.g., InstructGPT (Ouyang et al., 2022), Claude (Bai et al., 2022), and Gemini (Team et al., 2023)), but they assume transitive preferences and scalar reward functions. Recent empirical studies reveal that human preferences often exhibit non-transitive patterns and heterogeneous structures that violate these assumptions (Ethayarajh et al., 2024; Wu et al., 2024).

This has motivated *game-theoretic formulations of alignment* that treat preference optimization as finding Nash equilibria in games defined by general preference oracles (Munos et al., 2023). In the Nash learning from human feedback (NLHF) paradigm, a well-aligned policy constitutes an optimal strategy that competing policies cannot exploit, thereby achieving strategic optimality rather than mediocrity. Subsequent work has explored this paradigm through no-regret learning (Zhang et al., 2025b), optimistic mirror descent (Zhang et al., 2025a), and extragradient updates (Zhou et al., 2025), thereby advancing both theoretical guarantees and empirical stability relative to traditional RLHF.

---

$^{*}$Equal contribution.
$^{\dagger}$Corresponding authors.

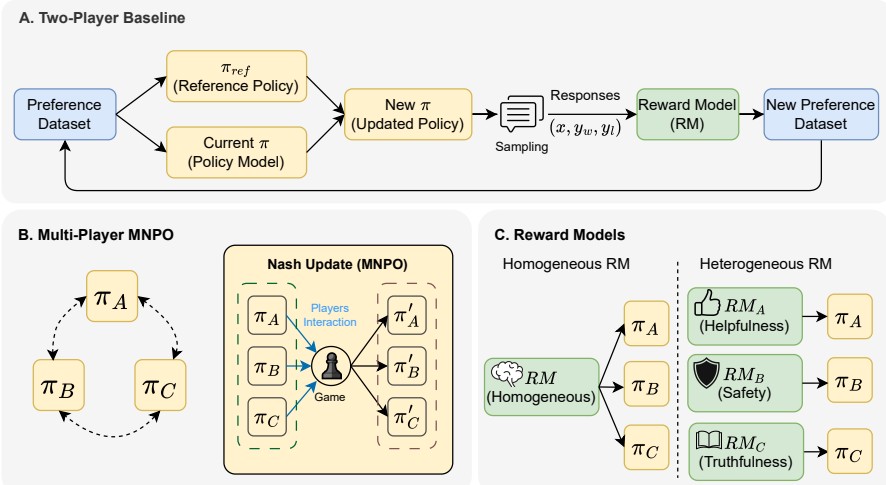

Figure 1: Overview of the two-player baseline and our multiplayer MNPO training paradigm.

Despite these advances, existing NLHF formulations remain constrained to *two-player settings*, where a single policy competes against one opponent. However, real-world preference alignment rarely resembles a two-agent interaction. Instead, it often involves a mixture of annotators, heterogeneous evaluation criteria, multiple reward models, or even a sequence of historical model checkpoints—creating inherently multi-source, and sometimes conflicting, preference signals (Freund & Schapire, 1999). Reducing this complex landscape to a single opponent introduces a *bottleneck*: the policy is optimized against only one distribution at a time, resulting in oscillatory behavior, narrow exploration, and a brittle approximation of the broader preference population.

These limitations highlight the need for a framework that explicitly models alignment as competition against an *entire population* rather than a single synthetic opponent. In this work, we show that extending to the multiplayer setting (Freund & Schapire, 1999) provides a principled mean-field approximation that reduces gradient variance, stabilizes optimization, and more precisely captures diverse preference structures. Crucially, when all policies share the same preference oracle—as naturally occurs when competing against historical versions of a single policy trajectory—the resulting symmetric game admits strong theoretical guarantees via multiplicative weights update.

We address these challenges by proposing *Multiplayer Nash Preference Optimization (MNPO)*, a principled framework that generalizes two-player preference optimization to $n$-player games. In MNPO, each policy simultaneously competes against a population of other policies while being regularized toward a reference model, producing a competitive equilibrium that balances performance against the population with adherence to a trusted baseline. Our contributions are threefold:

- **Theoretical Framework**: We establish that MNPO with homogeneous preference oracles admits natural equilibrium characterizations, including well-defined Nash policies and duality gaps that measure alignment quality. We prove that MNPO inherits the desirable convergence properties of existing two-player formulations while enabling richer equilibrium dynamics.

- **Algorithmic Innovation**: We introduce TD-MNPO, where opponent sets evolve adaptively using weighted combinations of historical policies, naturally yielding provable convergence guarantees. We further propose a heterogeneous extension (HT-MNPO) that, although it lacks formal guarantees, demonstrates strong empirical performance across diverse preference sources.

- **Empirical Validation**: Through comprehensive experiments on instruction-following and reasoning benchmarks, we demonstrate that MNPO consistently outperforms existing NLHF baselines, particularly excelling in scenarios involving diverse preferences and complex evaluation criteria.

Our analysis reveals that MNPO provides a unifying perspective on RLHF, subsuming many existing methods as special cases while offering improved robustness in multi-agent alignment scenarios. By bridging recent advances in NLHF with the practical demands of aligning LLMs with diverse, potentially non-transitive human preferences, MNPO establishes a scalable foundation for next-generation alignment techniques.

## 2 RLHF PRELIMINARIES

**Notation.** $x \in \mathcal{X}$ denotes a prompt and $\mathcal{X}$ is the prompt space. $x$ is assumed to be sampled from a fixed but unknown distribution $d_0$. An LLM is characterized by a policy $\pi : \mathcal{X} \to \Delta(\mathcal{Y})$ that takes a prompt $x$ as input and outputs a distribution over the response space $\mathcal{Y}$. The response $y \in \mathcal{Y}$ is then sampled from the distribution $\pi(\cdot \mid x)$.

**Bradley–Terry Model Assumption.** The prevalent RLHF framework (Christiano et al., 2017; Ouyang et al., 2022) assumes the Bradley–Terry model. It assumes that there exists a reward function $r^*$ such that for any $x \in \mathcal{X}$ and $y^1, y^2 \in \mathcal{Y}$, we have $\mathbb{P}\left(y^1 \succ y^2 \mid x\right) = \frac{\exp\left(r^*\left(x, y^1\right)\right)}{\exp(r^*(x, y^1)) + \exp(r^*(x, y^2))} = \sigma\left(r^*\left(x, y^1\right) - r^*\left(x, y^2\right)\right)$. After learning a reward function $R(\cdot, \cdot)$, RLHF algorithms aim to maximize the following KL-regularized objective with preference optimization RL algorithms such as PPO (Schulman et al., 2017):

$$J(\pi) = \mathbb{E}_{x \sim d_0}\left[\mathbb{E}_{y \sim \pi(\cdot \mid x)}[R(x, y)] - \tau \, \mathrm{KL}\left(\pi(\cdot \mid x) \| \pi_{\mathrm{ref}}(\cdot \mid x)\right)\right]. \tag{1}$$

Here $\pi_{\mathrm{ref}}$ is the reference policy, which is usually a supervised fine-tuned LLM, and $\tau > 0$ is the regularization parameter. By maximizing the objective, the obtained policy simultaneously achieves a high reward and stays close to $\pi_{\mathrm{ref}}$, which can mitigate reward hacking (Tien et al., 2022; Skalse et al., 2022) to some extent.

**General Preference Oracle.** The Bradley–Terry model assumption may not always hold in practice. Recent studies (Munos et al., 2023; Calandriello et al., 2024; Ye et al., 2024; Zhang et al., 2025b;a) have instead directly considered the general preference distribution $\mathbb{P}$ without imposing additional assumptions, framing the preference optimization problem as a two-player game. They assume the existence of a *preference oracle* $\mathbb{P} : \mathcal{X} \times \mathcal{Y} \times \mathcal{Y} \to [0, 1]$. It can be queried to obtain binary preference signals as $z \sim \mathrm{Ber}\left(\mathbb{P}\left(y^1 \succ y^2 \mid x\right)\right)$, where $z = 1$ indicates that $y^1$ is preferred to $y^2$, and $z = 0$ indicates the opposite. The preference distribution is introduced as $\lambda_{\mathbb{P}}(x, y, y') = (y, y') \cdot \mathbb{I}\left[U < \mathbb{P}\left(y \succ y' \mid x\right)\right] + (y', y) \cdot \mathbb{I}\left[U \geq \mathbb{P}\left(y \succ y' \mid x\right)\right]$, where $U \sim \mathrm{Uniform}(0, 1)$ is a random variable. Given two policies $\pi_1$ and $\pi_2$, LLMs are aligned using this general preference oracle, and the game objective is written as:

$$J\left(\pi_1, \pi_2\right) = \mathbb{E}_{x \sim d_0}\left[\mathbb{E}_{y_1 \sim \pi_1, y_2 \sim \pi_2}\left[\mathbb{P}\left(y_1 \succ y_2 \mid x\right)\right] - \tau \, \mathrm{KL}\left(\pi_1 \| \pi_{\mathrm{ref}}\right) + \tau \, \mathrm{KL}\left(\pi_2 \| \pi_{\mathrm{ref}}\right)\right] \tag{2}$$

where the max-player $\pi_1$ aims to maximize the objective, and the min-player $\pi_2$ aims to minimize the objective. The goal of both players is to maximize their winning rates against the opponent while not deviating too far from $\pi_{\mathrm{ref}}$, which shares a similar spirit with the objective of Eq. 1.

**Nash Policy and Duality Gap.** Without loss of generality, we restrict our attention to the policy class $\Pi$ containing the policies with the same support set as $\pi_{\mathrm{ref}}$. The Nash equilibrium of the game in Eq. 2 is then defined as:

$$\pi_1^*, \pi_2^* := \underset{\pi_1 \in \Pi}{\mathrm{argmax}} \, \underset{\pi_2 \in \Pi}{\mathrm{argmin}} \, J\left(\pi_1, \pi_2\right). \tag{3}$$

Due to the symmetry of the two players, their Nash policies are unique and coincide, meaning $\pi_1^* = \pi_2^* = \pi^*$ (Ye et al., 2024). Interestingly, $J\left(\pi^*, \pi\right) \geq 0.5$ always holds for $\forall \pi \in \Pi$, as $J\left(\pi^*, \pi^*\right) = 0.5$ indicates that $\pi^*$ is the best response against itself. To quantify how well a policy $\pi$ approximates the Nash policy $\pi^*$, we define the duality gap as:

$$\mathrm{DualGap}(\pi) := \max_{\pi_1} J\left(\pi_1, \pi\right) - \min_{\pi_2} J\left(\pi, \pi_2\right). \tag{4}$$

The duality gap is nonnegative and $\mathrm{DualGap}(\pi) = 0$ if and only if $\pi = \pi^*$. Hence, our goal is to find a policy that minimizes the duality gap. Once we achieve $\mathrm{DualGap}(\pi) \leq \epsilon$, we say that $\pi$ is an *$\epsilon$-approximate Nash policy*.

## 3 METHOD

### 3.1 RLHF AS MULTIPLAYER GAMES

To extend the two-player preference optimization objective to a multiplayer setting $\{\pi_i\}_{i=1}^n$, we consider a framework where each policy seeks to maximize its average preference probability against

all other policies while regularizing toward a reference policy. We first focus on the *homogeneous multiplayer setting* where all players share the same preference oracle. This symmetric structure underpins our theoretical guarantees.

**Plackett–Luce Reward Learning Assumption.** To generalize the maximum-likelihood reward learning objective for the Bradley–Terry model to one-vs-many comparisons, we adopt the *Plackett–Luce* framework. Specifically, we replace the pairwise logistic term $\log \sigma \left( R\left(x, y^1\right) - R\left(x, y^2\right) \right)$ with a *softmax* over multiple alternatives, extending the Bradley–Terry model to accommodate listwise comparisons. This Plackett–Luce model (Debreu, 1960; Plackett, 1975) maintains the interpretability of reward-based preferences while scaling to more complex decision-making scenarios. Formally, given a learned reward function $R(x, y)$ and a dataset $\mathcal{D}$ containing tuples $\left(x, \left\{y^1, y^2, \ldots, y^k\right\}\right)$, where $\left\{y^1, y^2, \ldots, y^k\right\}$ is a pool of $k$ to-be-ranked items, the probability that $y^i$ is preferred over the remaining pool of entities $\left\{y^j\right\}_{j \neq i}$ under the Plackett–Luce model is:

$$\mathbb{P}\left(y^i \succ \left\{y^j\right\}_{j \neq i} \mid x\right) = \frac{\exp(R\left(x, y^i\right))}{\exp(R\left(x, y^i\right)) + \sum_{j \neq i} \exp(R\left(x, y^j\right))}. \tag{5}$$

The corresponding negative log-likelihood for a single comparison is: $-\log \mathbb{P}\left(y^i \succ \left\{y^j\right\}_{j \neq i} \mid x\right) = \log \left( \exp\left(R\left(x, y^i\right)\right) + \sum_{j \neq i} \exp\left(R\left(x, y^j\right)\right)\right) - R\left(x, y^i\right)$. Consequently, the generalized reward learning objective for learning $R$ becomes:

$$\arg \max_{R \in \mathcal{R}} \mathbb{E}_{\left(x, \left\{y^{1:k}\right\}\right) \sim \mathcal{D}} \mathbb{E}_{y^i \in \left\{y^{1:k}\right\}} \underbrace{\left[ R\left(x, y^i\right) - \log \left( \exp\left(R\left(x, y^i\right)\right) + \sum_{j \neq i} \exp\left(R\left(x, y^j\right)\right)\right)\right]}_{\text{Per-comparison log-likelihood}}. \tag{6}$$

Several key observations arise from this formulation. First, when $k = 2$ for a one-vs-one comparison, Eq. 6 reduces to the vanilla Bradley–Terry objective, given by $\log \sigma \left(R\left(x, y\right) - R\left(x, y'\right)\right)$, since $\sigma(a) = \frac{e^a}{1+e^a}$. Second, this objective maximizes the gap between the reward of $y^i$ and the log-sum-exp (LSE) of all alternatives, effectively applying a soft maximum over competitors. This penalizes cases where $y^i$ fails to dominate the collective "strength" of the dispreferred items $\left\{y^j\right\}_{j \neq i}$.

**Homogeneous Multiplayer Preference Oracle.** We consider the case where all $n$ players share the same *universal preference oracle* $\mathbb{P}: \mathcal{X} \times \mathcal{Y} \times \{\mathcal{Y}\}^{n-1} \to [0, 1]$. It directly compares $y^i$ with a group of responses $\left\{y^j\right\}_{j \neq i}$ and outputs binary preference signals $z \sim \text{Ber}\left(\mathbb{P}\left(y^i \succ \left\{y^j\right\}_{j \neq i} \mid x\right)\right)$. Then, each policy $\pi_i$ competes against the other $n - 1$ players, and the objective function becomes:

$$J\left(\pi_i, \{\pi_j\}_{j \neq i}\right) = \mathbb{E}_{x \sim d_0} \left[ \mathbb{E}_{y^i \sim \pi_i, \{y^j \mid y^j \sim \pi_j\}_{j \neq i}} \left[ \mathbb{P}\left(y^i \succ \left\{y^j\right\}_{j \neq i} \mid x\right)\right] - \tau \, \text{KL}\left(\pi_i(\cdot \mid x) \| \pi_{\text{ref}}(\cdot \mid x)\right)\right]. \tag{7}$$

Here, each policy $\pi_i$ maximizes its expected preference against all other players $\{\pi_j\}_{j \neq i}$ while remaining close to the reference policy $\pi_{\text{ref}}$ via a KL penalty. The KL term, weighted by $\tau$, prevents over-optimization and ensures behavioral stability. All policies are updated concurrently: at each step, player $i$ improves its own objective $J$, leading the population toward a competitive equilibrium that balances performance against opponents and adherence to $\pi_{\text{ref}}$.

Eq. 7 exhibits several important properties. *(i)* Symmetric treatment: All policies are treated equally and compete in a symmetric manner, ensuring that $\pi_1^* = \pi_2^* = \cdots = \pi_n^*$ at equilibrium. *(ii)* Decentralized optimization: Each policy's update depends only on its own actions and the aggregate behavior of its opponents, thereby avoiding complex interdependencies. *(iii)* Generalization of the two-player case: When $n = 2$, each policy's objective reduces to maximizing its pairwise preference probability subject to its own KL penalty, as shown in Eq. 2.

**Nash Equilibrium and Duality Gap.** In this $n$-player game with homogeneous preference oracles, the Nash equilibrium is a policy $\pi^*$ where no player can improve their respective objectives by unilaterally deviating. Formally, for all $\pi_i \in \Pi$:

$$J\left(\pi_i^*, \{\pi_j^*\}_{j \neq i}\right) \geq J\left(\pi_i, \{\pi_j^*\}_{j \neq i}\right), \quad \forall i \in \{1, \ldots, n\}. \tag{8}$$

To quantify how far a given policy $\pi$ is from the Nash policy $\pi^*$, we define the *unified duality gap* in the multiplayer setting. For player $i$ with opponents fixed as $O_\pi = \{\pi_j\}_{j=1}^{n-1}$,

$$\text{DualGap}(\pi) = \max_{\pi' \in \Pi} J(\pi', O_\pi) - J(\pi, O_\pi). \tag{9}$$

This measures the maximum gain player $i$ could achieve by unilaterally switching to an alternative strategy while all opponents remain fixed. A Nash equilibrium is characterized by $\text{DualGap}(\pi^*) = 0$, and $\text{DualGap}(\pi) \leq \epsilon$ indicates that $\pi$ is an $\epsilon$-approximate Nash policy.

**Multiplayer Nash Preference Optimization.** There are well-known algorithms that approximately solve the Nash equilibrium in a constant-sum multiplayer game. In this work, we follow (Freund & Schapire, 1999) to establish an iterative framework that can asymptotically converge to the optimal policy on average. Given a learning rate $\eta$ of online mirror descent update, we start with a theoretical analysis that conceptually solves the homogeneous multiplayer game as follows [1]:

$$\pi_i^{(t+1)}(y \mid x) \propto \left( \prod_{j \neq i} \pi_j^{(t)}(y \mid x) \right)^{\frac{1}{n-1}} \exp\left( \frac{\eta}{n-1} \sum_{j \neq i} \mathbb{P}\left( y \succ \pi_j^{(t)} \mid x \right) \right). \tag{10}$$

This iterative framework starts from a base policy $\pi^{(0)}$ such as $\pi_{\text{ref}}$. In each iteration, the updated policy $\pi^{(t+1)}$ is obtained from the reference policy $\pi^{(t)}$ following the multiplicative weight update. Particularly, a response $y$ should have a higher probability weight if it has a higher average advantage over the current policy $\pi^{(t)}$. Notably, Eq. 10 provides convergence guarantees to a Nash equilibrium in this homogeneous circumstance, ensuring that the average policy $\bar{\pi}^{(T)} = \frac{1}{T} \sum_{t=1}^T \pi^{(t)}$ converges to an $\epsilon$-approximate Nash equilibrium with $\epsilon = O(1/\sqrt{T})$ regret bound (Hart & Mas-Colell, 2000).

Eq. 10 is equivalent to $\pi_i^{(t+1)}(y \mid x) = \prod_{j \neq i} \pi_j^{(t)}(y \mid x)^{\frac{1}{n-1}} \exp\left( \frac{\eta}{n-1} \left( y \succ \pi_j^{(t)} \mid x \right) \right) / Z_{\pi^{(t)}}(x)$, where $Z_{\pi^{(t)}}(x)$ is the normalization factor (a.k.a, the partition function). Then, for any fixed $x$ and $y$, each ideal update policy $\pi_i^{(t+1)}$ should satisfy:

$$\frac{1}{n-1} \sum_{j \neq i} \log \frac{\pi_i^{(t+1)}(y \mid x)}{\pi_j^{(t)}(y \mid x)} = \frac{\eta}{n-1} \sum_{j \neq i} \mathbb{P}\left( y \succ \pi_j^{(t)} \mid x \right) - \log Z_{\pi^{(t)}}. \tag{11}$$

Note that direct computation of $\pi^{(t+1)}$ involves a normalization factor, which is intractable for the exponentially large response space $\mathcal{Y}$. To avoid computing this normalization factor, we consider the logarithmic ratio between response pair $y$ and $y'$, and define the function $h_t(\pi, y, y')$ as:

$$h_t\left( \pi, y, y' \right) = \log \frac{\pi(y \mid x)}{\pi(y' \mid x)} - \frac{1}{n-1} \sum_{j \neq i} \left( \log \frac{\pi_j^{(t)}(y \mid x)}{\pi_j^{(t)}(y' \mid x)} \right). \tag{12}$$

From Eq. 11, we know that the following equality holds for any response pair $y, y' \in \text{Supp}(\pi_{\text{ref}})$:

$$h_t\left( \pi^{(t+1)}, y, y' \right) = \frac{\eta}{n-1} \sum_{j \neq i} \mathbb{P}\left( y \succ \pi_j^{(t)} \mid x \right) - \mathbb{P}\left( y' \succ \pi_j^{(t)} \mid x \right) \tag{13}$$

Based on this observation, we define the loss function $L_t(\pi)$ and update the policy $\pi^{(t+1)}$ as:

$$\pi^{(t+1)} \leftarrow \underset{\pi}{\arg\min} \; \mathbb{E}_{y_w, y_l \sim D_t} \underbrace{\left[ \left( h_t\left( \pi, y_w, y_l \right) - \frac{\eta}{n-1} \sum_{j \neq i} \mathbb{P}\left( y \succ \pi_j^{(t)} \right) - \mathbb{P}\left( y' \succ \pi_j^{(t)} \right) \right)^2 \right]}_{L_t(\pi)} \tag{14}$$

It is clear to see that $\pi^{(t+1)}$ is the minimizer of $L_t(\pi)$ since $L_t\left( \pi^{(t+1)} \right) = 0$. Furthermore, in the following lemma, we show that $\pi^{(t+1)}$ is the unique minimizer of $L_t$ within the policy class $\Pi$.

**Lemma 1.** *For each $t \in [T]$, $\pi_{t+1}$ in Eq. 14 is the unique minimizer of $L_t(\pi)$ within $\Pi$.*

---

[1] For notational conciseness and avoiding confusion, we use a superscript to denote time here, so that $\pi^{(t)}$ is equivalent to $\pi_t$.

The proof is deferred to Appendix F.1. Moreover, we replace the tricky term $\mathbb{P}\left(y \succ \pi_j^{(t)}\right)$ with a hyperparameter $\eta$ and propose the following loss to bypass it:

$$L'_t(\pi) = \mathbb{E}_{y,y' \sim \pi_t,\, y_w, y_l \sim \lambda_{\mathbb{P}}(y,y')} \left[ \left( h_t\left(\pi, y_w, y_l\right) - \frac{1}{2\eta} \right)^2 \right]. \tag{15}$$

**Proposition 1.** *For any policy $\pi \in \Pi$ and any iteration $t \in [T]$, $L'_t(\pi)$ is equivalent to $L_t(\pi)$, differing only by an additive constant that is independent of $\pi$.*

See the proof in Appendix F.2. Here, the response pair $(y, y')$ is directly sampled from the current policy $\pi^{(t)}$, which is crucial for the equivalence between $L'_t(\pi)$ and $L_t(\pi)$.

## 3.2 MULTIPLAYER NASH PREFERENCE OPTIMIZATION

**Reward-Enhanced MNPO.** While our framework is motivated by moving beyond the limitations of purely scalar reward-based approaches, this does not preclude the beneficial incorporation of reward information when available. The key distinction lies in how rewards are utilized: rather than relying solely on reward-based rankings with implicit transitivity assumptions (as in classical RLHF), MNPO can leverage reward information as auxiliary guidance while maintaining the flexibility to handle non-transitive preferences through its game-theoretic structure.

Reward-aware preference optimization (RPO) (Sun et al., 2025) demonstrates how *quantitative* reward signals can complement *qualitative* preference comparisons. It aligns learned implicit preference models with explicit reward models that provide graded assessments of response quality. This shift allows MNPO to move beyond binary preference margins and instead minimize discrepancies between the learned reward function $r_{\pi_\theta}(x, y)$ and the target reward model $r^\star(x, y)$. Concretely, RPO defines the loss over preference pairs as:

$$\mathcal{L}_{\mathrm{RPO}}^{\mathbb{D}}\left(\pi_\theta, \left(x, y^1, y^2\right) \mid r^\star, \pi_{\mathrm{ref}}, \beta, \eta\right) := \mathbb{D}\left[r_{\pi_\theta}\left(x, y^1\right) - r_{\pi_\theta}\left(x, y^2\right) \,\|\, \eta r^\star\left(x, y^1\right) - \eta r^\star\left(x, y^2\right)\right], \tag{16}$$

where $\left(x, y^1, y^2\right)$ represents a preference pair and the distance metric $\mathbb{D} : \mathbb{R} \times \mathbb{R} \to \mathbb{R}^*$ measures alignment between the model's implicit reward differences and the scaled reference reward differences. The hyperparameters $\eta \in \mathbb{R}^*$ and $\beta \in \mathbb{R}^*$ control the reward scale and regularization, respectively. This formulation directly encourages the policy $\pi_\theta$ to internalize human-aligned reward values, bridging qualitative preference optimization with quantitative reward modeling.

Importantly, the loss function Eq. 15 can be interpreted as a special case of RPO under a squared distance metric $\mathbb{D}^{\mathrm{sq}}$. Specifically, $L'_t(\pi)$ employs a learned reward model $r_{\pi_\theta}(x, y) := \mathbb{E}_{\pi_j}\left[\log \frac{\pi(y|x)}{\pi_j(y|x)}\right]$ and a reference reward model gap $\delta_{r^\star} := \eta\left(r^\star\left(x, y^1\right) - r^\star\left(x, y^2\right)\right) = \frac{1}{2\eta}$. This connection highlights that integrating reward awareness into multiplayer preference games enhances stability, interpretability, and alignment fidelity.

**Time-dependent Multiplayers** A central challenge in multiplayer optimization lies in defining and updating the set of opponent players $\left\{\pi_j^{(t)}\right\}_{j=1}^{n-1}$ in Eq. 13. Inspired by recent iterative preference optimization methods such as DNO (Rosset et al., 2024), SPIN (Chen et al., 2024), and INPO (Zhang et al., 2025b), which typically rely on past policy iterations (e.g., $\pi_{\mathrm{ref}}$ and $\pi^{(t-1)}$) to construct opponents, we adopt a time-dependent opponent selection mechanism.

At any iteration $t$, we construct the opponent set from a mixture of recent time-indexed historical policies $\{\pi_{t-j}\}_{j=0}^t$ ($n \le t + 1$), weighted by coefficients $\lambda_j$ with $\lambda_j \in [0, 1]$ and optionally $\sum_j \lambda_j \le 1$. The resulting time-dependent MNPO (TD-MNPO) loss is formulated as follows:

$$\mathcal{L}_{\mathrm{TD}}^{t,\mathbb{D}}(\pi \mid \beta, \{\lambda_j\}, \eta) = \mathbb{E}_{y,y' \sim \pi,\, y_w, y_l \sim \lambda_{\mathbb{P}}(y,y')} \mathbb{D}\left[\log \frac{\pi(y_w \mid x)}{\pi(y_l \mid x)} - \sum_{j=0}^{n-2} \lambda_j \log \frac{\pi_{t-j}(y_w \mid x)}{\pi_{t-j}(y_l \mid x)} \,\Big\|\, \eta \delta^\star\right], \tag{17}$$

where $\delta^\star$ encodes the target reward gap. By blending multiple past policies, this formulation stabilizes training, mitigates overfitting to transient fluctuations, and preserves temporal consistency.

Table 1: Time-dependent MNPO recovers many existing offline or online preference optimization algorithms. We denote the target reward gap as $\delta_{r^\star} := \eta \left( r^\star \left( x, y_1 \right) - r^\star \left( x, y_2 \right) \right)$. $\mathbb{D}^{\mathrm{sq}}$ and $\mathbb{D}^{\mathrm{bwd}}$ represent the squared distance and backward Bernoulli KL divergence, respectively.

| Algorithm | Num. Players | Opponents | Importance Weights | Dist. | Target Reward Gap |
|---|---|---|---|---|---|
| SimPO | $n = 1$ | – | – | $\mathbb{D}^{\mathrm{bwd}}$ | $\infty$ |
| CPO | $n = 1$ | – | – | $\mathbb{D}^{\mathrm{bwd}}$ | $\infty$ |
| DPO | $n = 2$ | $\pi_{\mathrm{ref}}$ | $\lambda_j = 1$ | $\mathbb{D}^{\mathrm{bwd}}$ | $\infty$ |
| Distill-DPO | $n = 2$ | $\pi_{\mathrm{ref}}$ | $\lambda_j = 1$ | $\mathbb{D}^{\mathrm{sq}}$ | $\infty$ |
| DNO | $n = 2$ | $\pi_t$ | $\lambda_j = 1$ | $\mathbb{D}^{\mathrm{bwd}}$ | $\infty$ |
| SPIN | $n = 2$ | $\pi_t$ | $\lambda_j = \beta$ | $\mathbb{D}^{\mathrm{bwd}}$ | $\infty$ |
| SPPO | $n = 2$ | $\pi_t$ | $\lambda_j = 1$ | $\mathbb{D}^{\mathrm{sq}}$ | $\eta \left( \widehat{P} \left( y \succ \pi_t \mid x \right) - \frac{1}{2} \right)$ |
| IPO | $n = 2$ | $\pi_{\mathrm{ref}}$ | $\lambda_j = 1$ | $\mathbb{D}^{\mathrm{sq}}$ | $\frac{1}{2\tau}$ |
| INPO | $n = 3$ | $\pi_t, \pi_{\mathrm{ref}}$ | $\lambda_j = \begin{cases} \frac{\tau}{\eta}, & \text{if } j = t \\ \frac{\eta - \tau}{\eta}, & \text{if } j = 1 \end{cases}$ | $\mathbb{D}^{\mathrm{sq}}$ | $\frac{1}{2\tau}$ |

**Connections to Existing RLHF.** This unified MNPO formulation in Eq. 17 reveals that many preference optimization algorithms can be recovered as special cases by varying the number of players $n$, choice of opponents $O_\pi$, distance metric $\mathbb{D}$, and target reward gap $\delta^\star$. For instance, DPO emerges by setting $n = 2$, $O_\pi = \pi_{\mathrm{ref}}$, and $\lambda_j = 1$. Table 1 summarizes these reductions, showing how time-dependent MNPO unifies offline and online preference optimization under one principled framework. We provide a broader overview of RLHF objectives in Appendix E.

Compared to static reference-based approaches, the reward-aware multiplayer formulation offers: *(i) Smoother policy evolution.* Unlike methods that rely solely on the most recent past policy, MNPO gradually incorporates multiple past policies, preventing abrupt shifts and stabilizing policy updates. *(ii) Greater robustness.* By combining historical opponents with a weighted mixture, MNPO mitigates the risk of overfitting to transient fluctuations in recent iterations. *(iii) Unified interpretation.* TD-MNPO seamlessly extends existing approaches into a unified formulation, allowing for flexible adaptation to different training scenarios. *(iv) Stable convergence.* The weighting scheme ensures that recent policies exert greater influence while preserving the broader learning trajectory, thereby leading to more stable convergence. By dynamically leveraging historical policies as opponent players, TD-MNPO enhances preference optimization, making it more adaptable and robust in evolving learning environments. The pseudo-algorithm is described in Appendix B.

### 3.3 MULTIPLAYER GAME WITH HETEROGENEOUS PREFERENCE ORACLES

**Multiplayers with Heterogeneous Preference Oracles.** In many real-world alignment scenarios, preference signals originate from multiple heterogeneous sources–such as annotators with different evaluation criteria or distinct reward models trained for separate dimensions of quality (e.g., helpfulness (Tan et al., 2025), safety (Dai et al., 2023), conciseness (Dumitru et al., 2025)). We therefore generalize MNPO to the heterogeneous setting by replacing the historical mixture over past policies with a mixture over opponent policies associated with distinct preference oracles.

Concretely, each player $\pi_i$ is paired with a distinct reward model $r_i(x, y)$, inducing a player-specific preference oracle $\mathbb{P}_i : \mathcal{X} \times \mathcal{Y} \times \{\mathcal{Y}\}^{n-1} \to [0, 1]$. The objective $J_i \left( \pi_i, \{\pi_j\}_{j \neq i} \right)$ becomes $\mathbb{E}_{x \sim d_0} \left[ \mathbb{E}_{y^i \sim \pi_i, y^j \sim \pi_j} \left[ \mathbb{P}_i \left( y^i \succ \{y^j\}_{j \neq i} \mid x \right) \right] - \tau \, \mathrm{KL} \left( \pi_i(\cdot \mid x) \| \pi_{\mathrm{ref}}(\cdot \mid x) \right) \right]$, which preserves the multiplayer structure while allowing each agent to learn under its own notion of preference.

**Game-Theoretic Properties.** When $\mathbb{P}_i \neq \mathbb{P}_j$, the resulting game is *general-sum* and lacks the symmetry needed for formal Nash equilibrium guarantees. In this setting, each player has its own objective $J_i$ with its own oracle

$$\mathbb{P}_i$$

, breaking the constant-sum structure that underpins the convergence guarantees of multiplicative weights update. Consequently, the iterative framework in Eq. 10 does not have formal convergence to the Nash equilibrium in the heterogeneous case (Daskalakis et al., 2009; Hart & Mas-Colell, 2000). To quantify the quality of a policy profile, we define a player-specific duality gap: $\mathrm{DualGap}_i(\pi_i) =$

$\max_{\pi_i' \in \Pi} J_i(\pi_i', O_\pi) - J_i(\pi_i, O_\pi)$, where $O_\pi = \{\pi_j\}_{j \neq i}$ are the fixed opponent policies. This measures player $i$'s incentive to deviate from the current strategy. We consider the policy profile $\{\pi_i\}_{i=1}^n$ to be near a stationary point when $\max_i \text{DualGap}_i(\pi_i) \leq \epsilon$, indicating that no player has a strong incentive to deviate unilaterally. Nevertheless, the algorithmic framework remains natural and principled: each policy optimizes with respect to the current opponent distribution using its own oracle and empirically yields effective solutions.

**Heterogeneous MNPO.** Let $\delta_i^\star$ denote the target reward gap induced by reward model $r_i$, following RPO's reward-aware interpretation. The heterogeneous MNPO (HT-MNPO) for player $i$ becomes:

$$\mathcal{L}_{\text{HT}}^{i,\mathbb{D}}(\pi_i \mid \beta, \{\lambda_j\}, \eta) = \mathbb{E}_{y,y' \sim \pi_i, y_w, y_l \sim \lambda_{\mathbb{P}_i}(y,y')} \mathbb{D}\left[\log \frac{\pi_i(y_w \mid x)}{\pi_i(y_l \mid x)} - \sum_{j \neq i} \lambda_j \log \frac{\pi_j(y_w \mid x)}{\pi_j(y_l \mid x)} \middle\| \eta \delta_i^\star \right], \quad (18)$$

where the mixture is over opponent policies $\{\pi_j\}_{j \neq i}$ and $\{\lambda_j\}$ are importance weights. Eq. 18 retains the equilibrium-seeking nature of MNPO, but each policy now internalizes a reward-gap signal specific to its own reward model. As a result, MNPO can align with heterogeneous or even conflicting evaluators, potentially yielding a population equilibrium that balances multiple dimensions of quality. As empirically demonstrated later, this heterogeneous formulation achieves strong performance in multi-reward-model scenarios, suggesting that it can find effective stationary points even without formal equilibrium guarantees. The pseudo-algorithm is illustrated in Appendix B.

## 4 EXPERIMENTAL SETUP

**Models and Training Settings.** We implement an online RLHF framework (Dong et al., 2024) with `Gemma-2-9B-it` (Team et al., 2024) as the base model. Our MNPO training consists of $T = 3$ iterations, where each iteration generates responses from the current policy on a fresh prompt set and updates the policy using preference feedback. To eliminate the need for costly human annotations, we employ the reward model `ArmoRM-Llama3-8B-v0.1` (Wang et al., 2024a) to provide preference signals for TD-MNPO. To simulate heterogeneous preference oracles, we select `Skywork-Reward-V2-Llama-3.1-8B` (Liu et al., 2025) and `Athene-RM-8B` (Frick et al., 2024) as additional reward models for HT-MNPO. Hyperparameter optimization is critical, as optimal configurations vary across base models and across iterations of the same model. Through empirical analysis, we find that maintaining $\beta$ within the range $[0.01, 10]$ consistently produces strong results. Furthermore, we observe that gradually increasing $\beta$ throughout training effectively mitigates training degradation while enabling continued model improvement. Complete implementation details and hyperparameter specifications are provided in Appendix C.

**Evaluation Benchmarks.** We evaluate primarily on three widely used open-ended instruction-following benchmarks: MT-Bench (Zheng et al., 2023), AlpacaEval 2 (Li et al., 2023), and Arena-Hard v0.1 (Li et al., 2024). As suggested by Dubois et al. (2024), we report the win rate (WR) for Arena-Hard and the length-controlled (LC) WR for AlpacaEval 2, as judged by GPT-5-mini rather than the outdated GPT-4 Turbo (Preview-1106).

Since RLHF alignment is known to sometimes degrade reasoning, calibration, and factual accuracy (Ouyang et al., 2022; Dong et al., 2024), we further assess performance on a broader set of 11 academic benchmarks. These benchmarks span multiple abilities, including explicit instruction following (Zhou et al., 2023), general knowledge (Clark et al., 2018; Rein et al., 2024; Hendrycks et al., 2020), commonsense reasoning (Sakaguchi et al., 2021; Lin et al., 2021; Zellers et al., 2019), and math/coding problem-solving (Lewkowycz et al., 2022; Chen et al., 2021).

In addition, we compare MNPO with a group of open-source LLMs, including `SmolLM3-3B` (Bakouch et al., 2025), `Llama-3.1-8B-it` (Dubey et al., 2024), `Olmo-2-32B-Instruct` (OLMo et al., 2025), `Tulu-2-DPO-70B`, `Llama-3.3-70B-it`, `Llama-3.1-Tulu-3-70B-DPO` (Lambert et al., 2025), `Mixtral-8x22B-it`, and `Qwen3-235B-it` (Yang et al., 2025), and closed-source LLMs such as `Gemini-2.5-Pro`, `GPT-5`, and `Claude-Sonnet-4`.

Table 2: Performance of various models on instruction-following and preference-alignment benchmarks (AlpacaEval 2.0, Arena-Hard, and MT-Bench), evaluated using GPT-5-mini as the judge.

| Model | Size | AlpacaEval 2.0 | Arena-Hard | MT-Bench |
|---|---|---|---|---|
| SFT Model | 9B | 50.15 | 44.97 | 6.49 |
| DPO | 9B | 54.35 | 45.63 | 6.87 |
| SimPO | 9B | 55.16 | 45.04 | 6.87 |
| SPPO | 9B | 55.97 | 43.89 | 6.86 |
| INPO | 9B | 56.09 | 48.03 | 6.95 |
| TD-MNPO | 9B | 57.27 | **52.26** | 7.03 |
| HT-MNPO (ArmoRM-Llama3) | 9B | 57.63 | 50.93 | **7.52** |
| HT-MNPO (Skywork-Reward-V2) | 9B | 56.01 | 50.34 | 7.40 |
| HT-MNPO (Athene-RM-8B) | 9B | **59.64** | 51.17 | 7.07 |
| SmolLM3-3B | 3B | 11.81 | 22.12 | 6.38 |
| LLaMA-3.1-8B-it | 8B | 5.24 | 39.16 | 5.37 |
| Olmo-2-32B-Instruct | 32B | 32.91 | 31.62 | 6.62 |
| Tulu-2-DPO-70B | 70B | 8.82 | 27.88 | 5.91 |
| Llama-3.1-Tulu-3-70B-DPO | 70B | 52.28 | 71.34 | 7.45 |
| LLaMA-3.3-70B-it | 70B | 29.44 | 59.21 | 7.73 |
| Mixtral-8x22B-it | 141B | 9.57 | 40.98 | 6.90 |
| Qwen3-235B-it | 235B | 84.97 | 88.71 | 8.27 |
| OpenAI/GPT-5 | - | 72.80 | 98.11 | 8.46 |
| Anthropic/Claude-Sonnet-4 | - | 62.24 | 77.58 | 8.26 |
| Google/Gemini-2.5-Pro | - | 90.93 | 86.98 | 7.78 |

Table 3: Model performance on instruction, knowledge, and commonsense benchmarks.

| Model | Instruction | Knowledge | | | Commonsense | | | AVG |
|---|---|---|---|---|---|---|---|---|
| | IFEval | GPQA | MMLU | ARC | HellaSwag | TruthfulQA | Winogrande | |
| SFT Model | 72.27 | 28.28 | 75.35 | 91.29 | 80.30 | 70.75 | 73.72 | 70.28 |
| DPO | 72.96 | 29.29 | 75.77 | 91.26 | 80.37 | 71.24 | 73.88 | 70.68 |
| SimPO | 73.79 | 32.32 | 76.79 | 91.09 | 78.90 | 63.40 | 72.93 | 69.60 |
| SPPO | 75.47 | 26.26 | 75.37 | 91.17 | 80.10 | 71.48 | 73.48 | 70.19 |
| INPO | 73.20 | 27.78 | 74.79 | 91.07 | 80.22 | 71.24 | 73.48 | 70.25 |
| TD-MNPO | 73.94 | 33.33 | 75.63 | 91.15 | 80.18 | 70.26 | 73.09 | 71.08 |
| HT-MNPO (ArmoRM-Llama3) | 75.05 | 29.80 | 75.60 | 91.23 | 80.29 | 70.38 | 73.48 | 70.83 |
| HT-MNPO (Skywork-Reward-V2) | 75.26 | 36.36 | 75.39 | 91.12 | 80.22 | 70.87 | 73.40 | **71.80** |
| HT-MNPO (Athene-RM-8B) | 74.42 | 30.81 | 75.40 | 91.18 | 80.44 | 71.36 | 73.32 | 70.99 |

# 5 EMPIRICAL RESULTS

**Instruction-Following and Preference Alignment.** Table 2 presents the performance of MNPO compared to existing preference optimization methods on three widely used instruction-following benchmarks: AlpacaEval 2.0 (Length-Controlled Win Rate, %), Arena-Hard (Win Rate, %), and MT-Bench (Score/10). All models were evaluated using `GPT-5-mini` as the judge. MNPO consistently outperforms all baseline methods across all three benchmarks. On AlpacaEval 2.0, MNPO achieves a score of 57.27, improving by 2.92 points over DPO (54.35), 2.11 points over SimPO (55.16), 1.30 points over SPPO (55.97), and 1.18 points over INPO (56.09). The improvements are even more pronounced on Arena-Hard, where MNPO scores 52.26, compared with the next-best method, INPO, at 48.03, representing a 4.23-point improvement. Notably, on this challenging benchmark, MNPO not only outperforms other preference optimization algorithms but also competes favorably with much larger open-source, fine-tuned models and even the latest closed-source models. It surpasses prominent models such as `Tulu-2-DPO (70B)` and `Mixtral-IT (141B)`. On MT-Bench, MNPO achieves 7.03, outperforming all baselines, with the closest competitor being INPO at 6.95. These results demonstrate that the multiplayer formulation in MNPO provides significant advantages for instruction-following tasks. The consistent improvements across all benchmarks suggest that the framework's ability to accommodate diverse preferences and non-transitive relationships yields better alignment with human expectations in open-ended generation tasks.

**Knowledge and Reasoning Capabilities.** Table 3 evaluates model performance on academic benchmarks covering instruction following, knowledge, and commonsense reasoning. The results show that MNPO maintains strong performance across diverse cognitive tasks while achieving preference alignment. MNPO achieves the highest average score of 71.08 across all benchmarks, outperforming

Table 4: Model performance on math and coding benchmarks.

| Model | Math | | | Code | AVG |
|---|---|---|---|---|---|
| | GSM8K | Minerva-Math | AIME-24 | HumanEval | |
| SFT Model | 81.96 | 44.12 | 0 | 60.37 | 46.61 |
| DPO | 82.03 | 45.96 | 0 | 59.76 | 46.94 |
| SimPO | 82.56 | 43.38 | 0 | 57.32 | 45.82 |
| SPPO | 82.11 | 47.43 | 0 | 59.76 | 47.33 |
| INPO | 82.94 | 46.32 | 0 | 59.15 | 47.10 |
| TD-MNPO | 82.64 | 44.85 | 3.33 | 61.59 | 48.10 |
| HT-MNPO (ArmoRM-Llama3) | 82.64 | 47.79 | 3.33 | 60.98 | **48.68** |
| HT-MNPO (Skywork-Reward-V2) | 82.03 | 49.63 | 0 | 59.76 | 47.86 |
| HT-MNPO (Athene-RM-8B) | 82.11 | 47.06 | 0 | 59.15 | 47.08 |

the SFT baseline (70.28) and all other preference optimization methods. Notably, MNPO achieves the best performance on GPQA (33.33), demonstrating strong graduate-level reasoning capabilities. The method also performs competitively on instruction following (IFEval: 73.94) and maintains solid performance on knowledge benchmarks such as MMLU (75.63) and on commonsense reasoning tasks. Importantly, unlike some preference optimization methods that show degradation on certain academic benchmarks (e.g., SimPO's drop to 63.40 on TruthfulQA), MNPO maintains relatively stable performance across all domains. This suggests that the multiplayer framework helps preserve the model's foundational capabilities while improving preference alignment.

**Mathematical and Coding Performance.**  Table 4 presents results on mathematical reasoning and coding benchmarks. MNPO achieves the highest average score of 48.10 across math and coding tasks, outperforming all baseline methods, including SPPO (47.33) and INPO (47.10). On the challenging AIME-24 benchmark, MNPO is the only method to achieve non-zero performance (3.33), while all other methods, including the SFT baseline, score 0. This demonstrates MNPO's superior capability in handling complex mathematical reasoning tasks. On HumanEval, MNPO achieves 61.59, representing the best coding performance among all methods. The results on GSM8K and Minerva-Math show that MNPO maintains competitive performance with existing methods while achieving superior results on the most challenging tasks. This pattern suggests that the multiplayer optimization framework is particularly beneficial for complex reasoning tasks that require handling multiple solution strategies.

## 6 RELATED WORK

**Reward-Model-Based RLHF.**  Classical RLHF trains a reward model on human preference data and optimizes a policy with KL-regularized policy gradients (e.g., PPO) (Christiano et al., 2017; Bai et al., 2022; Schulman et al., 2017). While effective, PPO-style updates can suffer from instability and high memory cost. To address this, GRPO removes the critic to improve training stability and memory efficiency at scale (e.g., DeepSeek-R1) (Shao et al., 2024; Guo et al., 2025), and DAPO further improves sample efficiency through dynamic sampling and refined loss objectives (Yu et al., 2025). VAPO shows that value learning can strengthen RLHF under appropriate design choices (Yuan et al., 2025). Nonetheless, optimizing against imperfect reward proxies remains vulnerable to reward hacking (Weng, 2024; Wen et al., 2024).

**RLHF with General Preferences.**  DPO bypasses reward modeling by directly optimizing a log-odds margin between preferred and dispreferred responses (Rafailov et al., 2023). This idea has inspired a family of extensions, including IPO (Azar et al., 2024), KTO (Ethayarajh et al., 2024), SimPO (Meng et al., 2024), and WPO (Zhou et al., 2024), which refine the constraint, scaling, or sampling strategy. Iterative and online forms introduce exploration and continual updates (Xiong et al., 2023; Dong et al., 2024; Xie et al., 2024), but most remain limited to static pairwise supervision.

## 7 CONCLUSION

We introduce Multiplayer Nash Preference Optimization, extending Nash learning from human feedback to multiplayer settings. Our framework admits or approximates well-defined Nash equilibria and

unifies existing preference optimization methods as special cases. Empirically, MNPO outperforms baselines across instruction-following, preference-alignment, and reasoning benchmarks. These results validate that multiplayer formulations better capture heterogeneous human preferences and provide more robust alignment for LLMs.

## ACKNOWLEDGMENT

This work was supported in part by RS-2024-00457882, National AI Research Lab Project.

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

## A    ADDITIONAL RELATED WORK ON GAME-THEORETIC RLHF

Another perspective frames preference optimization as a game between the model and its opponents, in which equilibrium-seeking methods yield stronger last-iterate guarantees. Self-play methods like SPIN (Chen et al., 2024), SPPO (Wu et al., 2024), and INPO (Zhang et al., 2025b) use no-regret dynamics, while Pre-DPO (Pan et al., 2025) and MPO (Wang et al., 2024b) adapt mirror descent. More recent methods, such as ONPO (Zhang et al., 2025a) and EGPO (Zhou et al., 2025), incorporate optimism and extragradient techniques to ensure stable convergence under noisy preferences. These advances primarily focus on two-player games but highlight the importance of equilibrium views in preference optimization. Extending RLHF to multiplayer interactions, as pursued by MNPO, generalizes beyond pairwise dynamics and enables richer equilibrium structures for alignment.

## B    PSEUDO-ALGORITHM OF MNPO

---

**Algorithm 1** Time-dependent Multiplayer Nash Preference Optimization (TD-MNPO)

---

1: **Require:** Number of iterations $T$, distance metric $\mathbb{D}$, weight coefficients $\{\lambda_j\}$, reward scaling parameter $\eta$, regularization parameter $\beta$, external policy $\{\pi_j\}$, reference policy $\pi_{\text{ref}}$, policy class $\Pi$, preference oracle $\mathbb{P}$.
2: **for** iteration $t = 1, 2, \ldots, T$ **do**
3:     Use current policy $\pi_t$ to generate response pairs $\{y_i^1, y_i^2\}_{i=1}^m$ where $y_i^1, y_i^2 \sim \pi_t$.
4:     Query the preference oracle $\mathbb{P}$ to get the preference dataset $D_t = \{y_i^+, y_i^-\}_{i=1}^m$.
5:     Calculate $\pi_{t+1}$ as: $\pi_{t+1} \leftarrow \underset{\pi \in \Pi}{\operatorname{argmin}} \, \mathcal{L}_{\text{TD-MNPO}}^{t,\mathbb{D}}$
6: **end for**
7: **Output** $\pi_{T+1}$

---

---

**Algorithm 2** Heterogeneous Multiplayer Nash Preference Optimization (HT-MNPO)

---

1: **Require:** Number of players $n$, iterations $T$, distance metric $\mathbb{D}$, importance weights $\{\lambda_j\}_{j=1}^n$, reward scaling parameter $\eta$, regularization parameter $\beta$, initial population of policies $\{\pi_i^{(1)}\}_{i=1}^n$, reference policy $\pi_{\text{ref}}$, policy class $\Pi$, heterogeneous preference oracles $\{\mathbb{P}_i\}_{i=1}^n$.
2: **for** iteration $t = 1, 2, \ldots, T$ **do**
3:     **for** player $i = 1, 2, \ldots, n$ **do**
4:         Use $\pi_i^{(t)}$ to generate response pairs $\{(y_{i,k}^1, y_{i,k}^2)\}_{k=1}^m$ where $y_{i,k}^1, y_{i,k}^2 \sim \pi_i^{(t)}(\cdot \mid x_k)$.
5:         Query the heterogeneous oracle $\mathbb{P}_i$ to obtain the dataset $D_t^{(i)} = \{(y_{i,k}^+, y_{i,k}^-)\}_{k=1}^m$.
6:         Update player $i$ by $\pi_i^{(t+1)} \leftarrow \underset{\pi \in \Pi}{\operatorname{argmin}} \, \mathcal{L}_{\text{HT-MNPO}}^{i,\mathbb{D}}$
7:     **end for**
8: **end for**
9: **Output:** $\left\{\pi_i^{(T+1)}\right\}_{i=1}^n$.

---

## C    EXPERIMENTAL DETAILS

**Hardware and Implementation**    All experiments are conducted on 8 NVIDIA H100 GPUs with 96GB memory. For baseline implementations, DPO (Rafailov et al., 2023) is trained using the official Hugging Face DPO Trainer, while SimPO (Meng et al., 2024)[2] and SPPO (Wu et al., 2024)[3] follow their official GitHub implementations. INPO (Zhang et al., 2025b) is reproduced according to the settings described in the paper.

---

[2] https://github.com/princeton-nlp/SimPO
[3] https://github.com/uclaml/SPPO

Table 5: Ablation on the number of players ($n$) in TD-MNPO, where we report AlpacaEval 2.0 (length-controlled win rate, %). Increasing $n$ consistently improves alignment quality, with diminishing returns beyond $n{=}3$.

| # Players ($n$) | 1 | 2 | 3 | 4 |
|---|---|---|---|---|
| **AlpacaEval 2.0** | 53.32 | 54.34 (+1.02) | 57.27 (+3.93) | **57.42** (+4.10) |

Table 6: Ablation on Different Base Models. AlpacaEval 2.0 (LC) results.

| Model | Llama-3-8B-it | INPO | TD-MNPO |
|---|---|---|---|
| **AlpacaEval 2.0** | 24.80 | 41.48 (+16.64) | **42.94** (+18.14) |

**Hyperparameters** For MNPO, we adopt hyperparameters consistent with SimPO and INPO, using a cosine learning-rate scheduler with a peak learning rate of $5 \times 10^{-7}$, a warmup ratio of 0.1, and a global batch size of 128. The optimizer is `AdamW` (Loshchilov & Hutter, 2017) without weight decay. We further perform a grid search for history weights at timesteps 1 and 2, selecting from $\{0, 0.1, 0.333, 0.5, 0.667, 0.9\}$. In addition, we perform a grid search for $\eta$ over $\{0.1, 0.01, 0.0075, 0.005, 0.002\}$ and set $\eta = 0.0075$.

**Training Data** For training data, we use Gemma2-Ultrafeedback-Armorm (Cui et al., 2023)[4], which contains approximately 60K training samples and 2K test samples. We retain both prompts and responses in iteration 1, while only prompts are used in iterations 2 and 3.

**Evaluation Framework** For evaluation, we adopt the EvalScope framework (Team, 2024)[5] (version 1.0.2) across all datasets in Tables 3 and 4, and also apply it for AlpacaEval 2.0 and Arena-Hard in Table 2. MT-Bench[6] evaluations are conducted following its official GitHub repository. The LLM judge is configured with `gpt5-mini-aug7-2025`, where the reasoning effort is set to "minimal," and all other parameters follow the default EvalScope settings.

## D  ADDITIONAL RESULTS

## E  FORMULATIONS OF PREFERENCE OPTIMIZATION OBJECTIVES

Table 9 provides a consolidated overview of various preference optimization objectives that have been proposed in the literature. Each method is presented in terms of its optimization objective given preference dataset $\mathcal{D}$ containing tuples $(x, y^+, y^-)$, where $x$ denotes the input prompt, and $y^+$ and $y^-$ denote the preferred (winning) and dispreferred (losing) responses, respectively. The table also specifies whether the method explicitly depends on a reference policy $\pi_{\text{ref}}$, whether it leverages the current policy $\pi_t$ during training, and whether the algorithm falls into the *offline* or *online* reinforcement learning regime.

## F  MATHEMATICAL ANALYSIS

### F.1  PROOF OF LEMMA 1

*Proof.* First, by construction $L_t(\pi^{(t+1)}) = 0$, hence $\pi^{(t+1)}$ is a minimizer. Suppose, for contradiction, there exists $\tilde{\pi} \in \Pi$ with $\tilde{\pi} \neq \pi^{(t+1)}$ and $L_t(\tilde{\pi}) = 0$. Then for every pair $y, y' \in \text{Supp}(\pi_{\text{ref}})$, we must have

$$h_t(\tilde{\pi}, y, y') \;=\; \frac{\eta}{n-1} \sum_{j \neq i} \Big( \mathbb{P}(y \succ \pi_j^{(t)} \mid x) - \mathbb{P}(y' \succ \pi_j^{(t)} \mid x) \Big).$$

---

[4] `https://huggingface.co/datasets/princeton-nlp/gemma2-ultrafeedback-armorm`
[5] `https://github.com/modelscope/evalscope`
[6] `https://github.com/lm-sys/FastChat/tree/main/fastchat/llm_judge`

Table 7: **Mean and standard deviation of model performance across three runs under different LLM judges.** This illustrates both stability and relative improvements across RLHF methods.

| Model | GPT-4-1106-preview | GPT-4.1 | GPT-5-mini |
|-------|--------------------|---------| -----------|
| SFT Model | $47.83 \pm 0.92$ | $41.95 \pm 0.13$ | $49.95 \pm 0.18$ |
| DPO | $48.32 \pm 1.26$ | $43.87 \pm 0.45$ | $53.98 \pm 0.34$ |
| INPO | $49.30 \pm 0.95$ | $47.78 \pm 0.14$ | $56.16 \pm 0.06$ |
| MNPO | $54.05 \pm 1.58$ | $49.21 \pm 0.14$ | $57.20 \pm 0.09$ |

Table 8: Ablation of 2-player vs. 3-player HT-MNPO on AlpacaEval 2.0. 2-player scores are averaged over all pairwise judge configurations for each reward model (e.g., Armo (Skywork) and Armo (Athene) for ArmoRM-Llama3), while 3-player results are copied from the full HT-MNPO setting in Tab. 2.

| Reward model | 2-player HT-MNPO | 3-player HT-MNPO | $\Delta$ |
|--------------|------------------|------------------|----------|
| ArmoRM-Llama3 | 55.43 | 57.63 | +2.20 |
| Skywork-Reward-V2 | 54.25 | 56.01 | +1.76 |
| Athene-RM-8B | 55.71 | 59.64 | +3.93 |

Unrolling $h_t$ and rearranging yields the pairwise ratio identity

$$\frac{\tilde{\pi}(y \mid x)}{\tilde{\pi}(y' \mid x)} = \frac{\exp\left(\frac{\eta}{n-1} \sum_{j \neq i} \mathbb{P}(y \succ \pi_j^{(t)} \mid x)\right) \prod_{j \neq i} \pi_j^{(t)}(y \mid x)^{\frac{1}{n-1}}}{\exp\left(\frac{\eta}{n-1} \sum_{j \neq i} \mathbb{P}(y' \succ \pi_j^{(t)} \mid x)\right) \prod_{j \neq i} \pi_j^{(t)}(y' \mid x)^{\frac{1}{n-1}}}.$$

Because $\text{Supp}(\tilde{\pi}) = \text{Supp}(\pi_{\text{ref}})$ and $\sum_y \tilde{\pi}(y \mid x) = 1$, these pairwise ratios determine $\tilde{\pi}(\cdot \mid x)$ uniquely—and that unique solution is exactly the normalized distribution implied by Eqs. 10 and 11, i.e., $\pi^{(t+1)}$. Hence $\tilde{\pi} = \pi^{(t+1)}$, a contradiction. Therefore, the minimizer is unique. □

### F.2 PROOF OF PROPOSITION 1

*Proof.* Introduce an indicator $I \sim \text{Ber}(\mathbb{P}(y \succ y' \mid x))$ for a pair $(y, y') \sim \pi^{(t)} \times \pi^{(t)}$. Consider

$$\widetilde{L}_t(\pi) = \mathbb{E}_{y, y' \sim \pi^{(t)}, I}\left(h_t(\pi, y, y') - \frac{I}{\eta}\right)^2.$$

Expanding and comparing $\widetilde{L}_t(\pi)$ with $L_t(\pi)$ shows the only difference lies in the cross-term $\mathbb{E}[h_t(\pi, y, y')(\mathbb{P}(y \succ \pi^{(t)} \mid x) - \mathbb{P}(y' \succ \pi^{(t)} \mid x))]$ versus $\mathbb{E}[h_t(\pi, y, y') I]$. One verifies these are equal by writing $h_t$ as a linear form in $\log \pi$, $\log \pi_j^{(t)}$ and using that $y$ and $y'$ are i.i.d. draws from $\pi^{(t)}$:

$$\mathbb{E}_{y, y'}[h_t(\pi, y, y')(\mathbb{P}(y \succ \pi^{(t)} \mid x) - \mathbb{P}(y' \succ \pi^{(t)} \mid x))] = \mathbb{E}_{y, y', I}[h_t(\pi, y, y') I].$$

(See INPO Appendix A.5 for the algebraic steps; the same symmetry argument applies verbatim.) Now expand $L'_t(\pi)$ by conditioning on $(y, y')$ and the preference sampler $\lambda_{\mathbb{P}}(y, y')$:

$$L'_t(\pi) = \mathbb{E}_{y, y'}\left[\mathbb{P}(y \succ y' \mid x)\left(h_t(\pi, y, y') - \frac{1}{2\eta}\right)^2 + \left(1 - \mathbb{P}(y \succ y' \mid x)\right)\left(h_t(\pi, y', y) - \frac{1}{2\eta}\right)^2\right].$$

Using $h_t(\pi, y', y) = -h_t(\pi, y, y')$ and completing the square shows $L'_t(\pi) = \widetilde{L}_t(\pi) + \text{const}$, where the constant depends only on the distribution of $(y, y')$ and $\mathbb{P}(\cdot)$, not on $\pi$. Combining with the equality of cross-terms above gives $L'_t(\pi) = L_t(\pi) + \text{const}$, as claimed. □

### F.3 THEORETICAL ANALYSIS OF EQ. 10

Here, we provide a theoretical analysis of the iterative update in Eq. 10:

$$\pi_i^{(t+1)}(y \mid x) \propto \left(\prod_{j \neq i} \pi_j^{(t)}(y \mid x)\right)^{\frac{1}{n-1}} \exp\left(\frac{\eta}{n-1} \sum_{j \neq i} \mathbb{P}\left(y \succ \pi_j^{(t)} \mid x\right)\right). \tag{19}$$

Table 9: Various preference optimization objectives given preference data $\mathcal{D} = (x, y^+, y^-)$, where $x$ is an input, and $y^+$ and $y^-$ are the winning and losing responses. $f$ is a class of divergence functions. $\Gamma(x, y)$ is the uncertainty estimator. $l$ is a convex decreasing loss function. $\widehat{P}(y \succ \pi_t \mid x)$ is the win rate over the distribution estimated by the average win rate over all the sampled responses $y_{1:K} \sim \pi_t(\cdot \mid x)$.

| Method | Objective | $\pi_{\text{ref}}$ | $\pi_t$ | RL Type |
|---|---|---|---|---|
| DPO (Rafailov et al., 2023) | $\mathbb{E}_{(x,y^+,y^-)\sim D} - \log \sigma \left( \beta \log \frac{\pi_\theta(y^+\mid x)}{\pi_{\text{ref}}(y^+\mid x)} - \beta \log \frac{\pi_\theta(y^-\mid x)}{\pi_{\text{ref}}(y^-\mid x)} \right)$ | ✓ | ✗ | Offline |
| $f$-DPO (Wang et al., 2023) | $\mathbb{E}_{(x,y^+,y^-)\sim D} - \log \sigma \left( \beta f' \left( \frac{\pi_\theta(y^+\mid x)}{\pi_{\text{ref}}(y^+\mid x)} \right) - \beta f' \left( \frac{\pi_\theta(y^-\mid x)}{\pi_{\text{ref}}(y^-\mid x)} \right) \right)$ | ✓ | ✗ | Offline |
| R-DPO (Park et al., 2024) | $\mathbb{E}_{(x,y^+,y^-)\sim D} - \log \sigma \left( \beta \log \frac{\pi_\theta(y^+\mid x)}{\pi_{\text{ref}}(y^+\mid x)} - \beta \log \frac{\pi_\theta(y^-\mid x)}{\pi_{\text{ref}}(y^-\mid x)} + (\alpha \mid y^+ \mid - \alpha \mid y^- \mid) \right)$ | ✓ | ✗ | Offline |
| Distill-DPO (Fisch et al., 2024) | $\mathbb{E}_{(x,y^+,y^-)\sim D} \left[ \log \frac{\pi_\theta(y^+\mid x)}{\pi_{\text{ref}}(y^+\mid x)} - \log \frac{\pi_\theta(y^-\mid x)}{\pi_{\text{ref}}(y^-\mid x)} - (r^*(x,y^+) - r^*(x,y^-)) \right]^2$ | ✓ | ✗ | Offline |
| GSHF (Xiong et al., 2023) | $\mathbb{E}_{(x,y^+,y^-)\sim D} - \log \sigma \left( \beta \log \frac{\pi_\theta(y^+\mid x)}{\pi_{\text{ref}}(y^+\mid x)} - \beta \log \frac{\pi_\theta(y^-\mid x)}{\pi_{\text{ref}}(y^-\mid x)} \right) + (\Gamma(x,y^+) - \Gamma(x,y^-))$ | ✓ | ✗ | Offline |
| KTO (Ethayarajh et al., 2024) | $\mathbb{E}_{(x,y^+,y^-)\sim D} - \lambda_w \sigma \left( \beta \log \frac{\pi_\theta(y^+\mid x)}{\pi_{\text{ref}}(y^+\mid x)} - z_{\text{ref}} \right) + \lambda_l \sigma \left( z_{\text{ref}} - \beta \log \frac{\pi_\theta(y^-\mid x)}{\pi_{\text{ref}}(y^-\mid x)} \right),$ where $z_{\text{ref}} = \mathbb{E}_{(x,y)\sim\mathcal{D}} [\beta \text{KL}(\pi_\theta(y\mid x) \| \pi_{\text{ref}}(y\mid x))]$ | ✓ | ✗ | Offline |
| IPO (Azar et al., 2024) | $\mathbb{E}_{(x,y^+,y^-)\sim D} \left[ \log \frac{\pi_\theta(y^+\mid x)}{\pi_{\text{ref}}(y^+\mid x)} - \log \frac{\pi_\theta(y^-\mid x)}{\pi_{\text{ref}}(y^-\mid x)} - \frac{1}{2\tau} \right]^2$ | ✓ | ✗ | Offline |
| SLiC-HF (Zhao et al., 2023) | $\mathbb{E}_{(x,y^+,y^-)\sim D} \max(0, \delta - \log \pi_\theta(y^+\mid x) + \log \pi_\theta(y^-\mid x)) - \lambda \log \pi_\theta(y^+\mid x)$ | ✗ | ✗ | Offline |
| RRHF (Yuan et al., 2023) | $\mathbb{E}_{(x,y^+,y^-)\sim D} \max \left( 0, -\frac{1}{\mid y^+\mid} \log \pi_\theta(y^+\mid x) + \frac{1}{\mid y^-\mid} \log \pi_\theta(y^-\mid x) \right) - \lambda \log \pi_\theta(y^+\mid x)$ | ✗ | ✗ | Offline |
| SimPO (Meng et al., 2024) | $\mathbb{E}_{(x,y^+,y^-)\sim D} - \log \sigma \left( \frac{\beta}{\mid y^+\mid} \log \pi_\theta(y^+\mid x) - \frac{\beta}{\mid y^-\mid} \log \pi_\theta(y^-\mid x) - \gamma \right)$ | ✗ | ✗ | Offline |
| CPO (Xu et al., 2024) | $\mathbb{E}_{(x,y^+,y^-)\sim D} - \log \sigma (\beta \log \pi_\theta(y^+\mid x) - \beta \log \pi_\theta(y^-\mid x)) - \lambda \log \pi_\theta(y^+\mid x)$ | ✗ | ✗ | Offline |
| ORPO (Hong et al., 2024) | $\mathbb{E}_{(x,y^+,y^-)\sim D} - \log p_\theta(y^+\mid x) - \lambda \log \sigma \left( \log \frac{p_\theta(y^+\mid x)}{1-p_\theta(y^+\mid x)} - \log \frac{p_\theta(y^-\mid x)}{1-p_\theta(y^-\mid x)} \right),$ where $p_\theta(y\mid x) = \exp \left( \frac{1}{\mid y\mid} \log \pi_\theta(y\mid x) \right)$ | ✗ | ✗ | Offline |
| DNO (Rosset et al., 2024) | $\mathbb{E}_{(x,y'_t,y''_t)\sim\mathcal{D}_t} - \sigma(r_t(x,y'_t) - r_t(x,y''_t)) \log \left[ \sigma \left( \eta \log \frac{\pi(y'_t\mid x)}{\pi_t(y'_t\mid x)} - \eta \log \frac{\pi(y''_t\mid x)}{\pi_t(y''_t\mid x)} \right) \right]$ $- \sigma(r_t(x,y''_t) - r_t(x,y'_t)) \log \left[ \sigma \left( \eta \log \frac{\pi(y''_t\mid x)}{\pi_t(y''_t\mid x)} - \eta \log \frac{\pi(y'_t\mid x)}{\pi_t(y'_t\mid x)} \right) \right]$ | ✗ | ✓ | Online |
| DNO-Prct (Rosset et al., 2024) | $\mathbb{E}_{(x,y^+_t,y^-_t)\sim\mathcal{D}_t} - \log \left[ \sigma \left( \widetilde{\eta} \log \frac{\pi(y^+_t\mid x)}{\pi_t(y^+_t\mid x)} - \widetilde{\eta} \log \frac{\pi(y^-_t\mid x)}{\pi_t(y^-_t\mid x)} \right) \right]$ | ✗ | ✓ | Online |
| SPIN (Chen et al., 2024) | $\mathbb{E}_{(x,y,y^-_t)\sim\mathcal{D}_t} - \ell \left( \beta \log \frac{\pi_\theta(y\mid x)}{\pi_t(y\mid x)} - \beta \log \frac{\pi_\theta(y'\mid x)}{\pi_t(y'\mid x)} \right)$ | ✗ | ✓ | Online |
| SPPO (Wu et al., 2024) | $\mathbb{E}_{(x,y,\widehat{P}(y\succ\pi_t\mid x))\sim\mathcal{D}_t} - \left[ \log \frac{\pi_\theta(y\mid x)}{\pi_t(y\mid x)} - \eta \left( \widehat{P}(y \succ \pi_t \mid x) - \frac{1}{2} \right) \right]^2$ | ✗ | ✓ | Online |
| INPO (Zhang et al., 2025b) | $\mathbb{E}_{(x,y^+_t,y^-_t)\sim D_t} - \left[ \frac{\tau}{\eta} \left( \log \frac{\pi_\theta(y^+_t)}{\pi_{\text{ref}}(y^+_t)} - \log \frac{\pi_\theta(y^-_t)}{\pi_{\text{ref}}(y^-_t)} \right) + \frac{\eta-\tau}{\eta} \left( \log \frac{\pi_\theta(y^+_t)}{\pi_t(y^+_t)} - \log \frac{\pi_\theta(y^-_t)}{\pi_t(y^-_t)} \right) - \frac{1}{2\tau} \right]^2$ | ✓ | ✓ | Online |
| ONPO (Zhang et al., 2025a) | $\mathbb{E}_{(x,y^+_t,y^-_t)\sim D_t} \left[ \log \frac{\pi_\theta(y^+_t)}{\pi'_t(y^+_t)} - \log \frac{\pi_\theta(y^-_t)}{\pi'_t(y^-_t)} - \frac{\eta}{2} \right]^2,$ where $\pi'_t = \arg\min_\pi \mathbb{E}_{(x,y^+_t,y^-_t)\sim D_t} \left[ \log \frac{\pi(y^+_t)}{\pi_t(y^+_t)} - \log \frac{\pi(y^-_t)}{\pi_t(y^-_t)} - \frac{\eta}{2} \right]^2$ | ✗ | ✓ | Online |
| MIO (Lv et al., 2025) | $\mathbb{E}_{(x,y^+,y^-)\sim D} \left[ \log \left( 1 + \frac{\pi_{\text{ref}}(y^+\mid x)}{\pi_\theta(y^+\mid x)} \right) + \frac{1}{2} \log \left( 1 + \frac{\pi_\theta(y^+\mid x)}{\pi_{\text{ref}}(y^+\mid x)} \right) + \frac{1}{2} \log \left( 1 + \frac{\pi_\theta(y^-\mid x)}{\pi_{\text{ref}}(y^-\mid x)} \right) \right]$ | ✓ | ✓ | Offline |

**Derivation from Mirror Descent.** Eq. 19 can be viewed as an instance of online mirror descent (OMD) with the KL divergence as the Bregman potential. At each step, player $i$ seeks to maximize the expected win probability against the population $\left\{ \pi_j^{(t)} \right\}_{j \neq i}$ subject to a KL regularization toward the opponent mixture:

$$\pi_i^{(t+1)} = \arg\max_{\pi \in \Pi} \frac{1}{n-1} \sum_{j \neq i} \left\langle \pi, P(\cdot \succ \pi_j^{(t)} \mid x) \right\rangle - \frac{1}{\eta} \text{KL} \left( \pi \left\| \left( \prod_{j \neq i} \pi_j^{(t)} \right)^{\frac{1}{n-1}} \right. \right).$$

Taking first-order conditions yields exactly the multiplicative-weights update in Eq. 19. Thus, the MNPO update inherits the regret guarantees of OMD: the average regret after $T$ rounds scales as $\mathcal{O}(1/\sqrt{T})$, ensuring convergence to equilibrium in the no-regret learning sense.

**Pairwise Ratio Dynamics.** To avoid computing the intractable partition function, we analyze the pairwise log-ratio

$$h_t(\pi, y, y') \;=\; \log \frac{\pi(y \mid x)}{\pi(y' \mid x)} - \frac{1}{n-1} \sum_{j \neq i} \log \frac{\pi_j^{(t)}(y \mid x)}{\pi_j^{(t)}(y' \mid x)}.$$

Eq. 11 in the main text shows that at the fixed point $\pi^{(t+1)}$, these ratios satisfy

$$h_t\left(\pi^{(t+1)}, y, y'\right) \;=\; \frac{\eta}{n-1} \sum_{j \neq i} \left(P(y \succ \pi_j^{(t)} \mid x) - P(y' \succ \pi_j^{(t)} \mid x)\right).$$

Hence, the log-ratio dynamics correspond to a consistent linearization of the preference margins across all opponents, ensuring that the update increases probability mass on responses with strictly higher average advantage.

**Equilibrium Properties.** By standard arguments in no-regret game dynamics (Freund & Schapire, 1999), if all players update according to Eq. 19, the joint empirical distribution converges to an $n$-player Nash equilibrium. Moreover, because the update is multiplicative in form, probabilities remain strictly positive on the support of $\pi_{\mathrm{ref}}$, preventing premature collapse.

**Interpretation.** Eq. 19 admits two complementary interpretations:

- *Population averaging:* the geometric mean term aggregates beliefs of all opponents, ensuring stability against heterogeneous policies.
- *Advantage weighting:* the exponential term amplifies responses that consistently outperform others, with learning rate $\eta$ controlling the exploration–exploitation trade-off.

Together, these properties explain why MNPO achieves robust convergence in multiplayer preference optimization while generalizing the two-player INPO update.

## G  LIMITATIONS AND FUTURE WORK

Like other algorithms in the RLHF paradigm, MNPO's performance is fundamentally linked to the quality of its preference data. Three primary limitations warrant consideration for future work.

First, the fidelity of the preference oracle serves as a performance ceiling. As the policy model improves and its generations become consistently high-quality, distinguishing between chosen and rejected responses becomes increasingly difficult for the preference oracle in practice. This diminishing discriminative capability can become a bottleneck for further improvement.

Second, the paradigm of simply increasing the probability of chosen responses and decreasing that of rejected ones may face diminishing returns as the preference gap narrows. When rejected responses are themselves of high quality, the binary preference signal becomes less informative, potentially slowing down or stalling the convergence of the policy model. Future research could explore more nuanced feedback mechanisms to address learning in this high-performance regime.

Third, our theoretical analysis focuses on the homogeneous multiplayer setting where all players share the same preference oracle. This restriction is necessary to ensure the game has a constant-sum structure and that multiplicative weight updates converge to a Nash equilibrium. The heterogeneous extension (HT-MNPO), while empirically effective, operates in a general-sum game where formal convergence guarantees do not apply. Future work could explore alternative equilibrium concepts (e.g., coarse correlated equilibrium) or game structures that provide theoretical grounding for heterogeneous preference optimization.

### G.1  EXTERNAL OPPONENT PLAYERS

**Formulation.** An alternative to using past-time policies as opponents is leveraging external LLMs to introduce a broader range of competitive dynamics. Given a set of $n-1$ LLM policies $\{\pi_j\}_{j=1}^{n-1}$, these opponent models can come from different sources: they may belong to distinct model families,

have varying parameter scales within the same architecture, be trained on different data distributions, or specialize in different domains. The external opponent MNPO (EO-MNPO) loss in this case is defined as:

$$\mathcal{L}_{\text{EO-MNPO}}^{t,\mathbb{D}}(\pi \mid \beta, \{\lambda_j\}, \eta) = \mathbb{D}\left[\sum_j \lambda_j \left(\log \frac{\pi(y \mid x)}{\pi_j(y \mid x)} - \log \frac{\pi(y' \mid x)}{\pi_j(y' \mid x)}\right)\bigg| \eta \delta_{r^\star}\right]. \quad (20)$$

Here, the constraint $\sum_j \lambda_j = 1$ ensures that the contributions of different opponent policies are appropriately weighted. This formulation introduces a key advantage: it enables preference optimization across diverse knowledge sources rather than being restricted to a single training trajectory.

To better interpret this formulation, we can draw parallels to knowledge distillation. Specifically, consider a scenario where we have a collection of teacher models $\{\pi_i^{\text{teacher}}\}$ and seek to refine an updated policy $\pi$ that remains close to both these teacher models and the original reference model $\pi_{\text{ref}}$. The objective function can be expressed as:

$$J(\pi) = \mathbb{E}_{x \sim d_0}\left[\mathbb{E}_{y \sim \pi}[R(x,y)] - \tau_0 \, \text{KL}\left(\pi\|\pi_{\text{ref}}\right) - \sum_i \tau_i \, \text{KL}\left(\pi\|\pi_i^{\text{teacher}}\right)\right]. \quad (21)$$

This formulation shows that MNPO with external opponent players can be viewed as an extension of knowledge distillation (Aminian et al., 2025), in which the learned policy integrates information from multiple expert models while balancing divergence from the reference model. Using the same derivation technique as DPO, we can recover Eq. 20, linking preference optimization to a generalized knowledge alignment framework.

**Proposition 2.** *The optimal solution to the maximum reward objective in Eq. 21 is equivalent to the learned reward model in Eq. 20.*

It is straightforward to show that the solution to the maximum reward objective in Eq. 21 takes the form: $\pi^*(y \mid x) = \frac{1}{Z(x)}\exp(R(x,y)/\tau)\pi_{\text{ref}}(y \mid x)^{\tau_0/\tau} \prod_i \pi_i^{\text{teacher}}(y \mid x)^{\tau_i/\tau}$, where $Z(x)$ is the partition function and $\tau = \tau_0 + \sum_i \tau_i$.

**Analysis.** EO-MNPO gains several benefits. First, unlike the self-comparison TD-MNPO, using a diverse pool of external LLMs exposes the policy to a broader range of feedback, leading to more robust optimization. Second, leveraging domain-specific expert models allows MNPO to fine-tune policies for specialized applications, improving generalization. Third, this framework aligns with broader trends in multi-agent RL, in which agents iteratively refine their strategies against multiple opponents, thereby enabling more sophisticated decision-making. By considering a dynamic set of external LLMs as opponent players, EO-MNPO extends beyond self-referential optimization, making it a more flexible and generalizable approach for preference optimization in LLMs.

