# OpenReview forum: "Multiplayer Nash Preference Optimization"
_ICLR.cc/2026/Conference — ICLR 2026 Oral_

### Official Review · Reviewer_RaDU · 2025-10-20

**Soundness:** 3
**Presentation:** 3
**Contribution:** 3
**Rating:** 6
**Confidence:** 4

**Summary:**

The paper extends the field of reinforcement learning from human feedback and alignment of LLMs.
Their primary assumption is that reward models are limited in expressing the complexity and diversity of human preferences therefore more general preferences models should be considered. In particular, they model n-wise comparisons instead of pairwise comparisons when one completions out of a pool of n is chosen as the best response by the LLM for a given prompt.
Since this is a non-transitive model, standard reward models can't be used and the authors formulate the problem as a game between n players and aim to approximate the Nash Equilibrium. However, they model generalises to settings when reward models are adequate choice for human preferences.
The proposed algorithm, MNPO, builds on online mirror descent that provides on average convergence guarantee and follow the previous line of work on Nash Learning from Human Feedback (Munos et al. 2023) and extends it to time-dependent opponent selection (TD-MNPO).
Their experimental results show consistently better performance than comparable algorithms and results on par with open-weight and closed-source models on standard benchmarks.

**Strengths:**

The main strength of the paper lies in its generality as highlighted in Table 1 furthermore in the time-dependent opponent selection that extends on the standard OMD approach.
The experimental results look promising and the comprehensive evaluation on a wide range of benchmarks is appreciated, however, I have some questions as discussed below.

**Weaknesses:**

I found the following points unclear and potential weaknesses of the paper:
1. On top of page 2, the authors motivate the multi-player formulation via diverse annotators and heterogeneous evaluation criteria that are parts of the feedback signal that evaluates multiple completions for a single prompt, however, the MNPO framework focuses on multiple models generating the responses. In my opinion, this motivation is more suitable to argue that preference modelling should be improved and not that alignment algorithms need to compare more than 2 responses. It would be great if the authors could clarify the connection between the motivation and the framework.
2. While the authors claim that the iterative framework they build upon has an asymptotic convergence guarantee to the optimal policy on average (Line 215-216), it is not discussed whether this guarantee holds in their case after introducing modifications to the online mirror descent in Eq. 15 and 16 especially as it moves from an offline dataset $D_t$ in Eq. (15) to an online optimisation problem in Eq. (16). Furthermore, how it carries over to the TD-MNPO algorithm. The last paragraph in Section 3 discussed the benefits of TD-MNPO without theoretical or empirical validation. Discussion on this would be much appreciated.
3. TD-MNPO requires a mixture of all previous policies of opponents in its loss function defined in Eq. 18. This can be limiting when either n or T are large. I suggest the authors to comment on how this can be scaled to large n or T.

**Questions:**

I have the following questions to the authors that would be great to clarify beyond the points mentioned in the Weaknesses section
1. In the definition of the DualGap for multiplayer games, Eq. (10), the first term takes both the expectation and the maximum over $\pi_j$. Could the authors clarify this notation and definition?
2. In the experiments, Gemma-2-9B-it was used as a base model which is an older version and has already been preference fine-tuned. Why did the authors choose this model and do the results hold when MNPO is initialised with an on SFT model? Due to this, the base model already achieves high scores on most benchmarks leaving small room for improvement and the differences between algorithms and models are marginal.
3. How does the model compare against more recent open-source LLMs e.g. Tulu3 instead of Tulu2, Olmo2, or SmolLM3? Some models for comparison seems outdated and would be better to compare against more recent variants.
4. Does the results carry over more general preference oracle? Current experiments use a reward model, ArmoRM-Llama3-8B-v0.1, which results in transitive preference signals while game-theoretic approaches are meant to optimise preference signals that are more diverse.
5. Why did the authors choose to use GPT-5 as judge for the benchmarks? It is not standard for some of them, e.g., AlpacaEval 2.0 and its alignment with the human annotators is not evaluated, therefore, results with these models are less reliable. I would suggest to validate results with some of the standard annotators provided by the original codebase.
6. How did the authors decide that 3 iterations for MNPO is sufficient? Results and comments on convergence and DualGap would be appreciated to judge whether the model really converged.

---

> ### Author Response · Authors · 2025-11-25
> **Response to Reviewer RaDU**
>
> ## \[W1] Motivation: diverse annotators vs. multiple models
>
> > On top of page 2, the authors motivate the multi-player formulation via diverse annotators and heterogeneous evaluation criteria that are parts of the feedback signal that evaluates multiple completions for a single prompt, however, the MNPO framework focuses on multiple models generating the responses. In my opinion, this motivation is more suitable to argue that preference modelling should be improved and not that alignment algorithms need to compare more than 2 responses. It would be great if the authors could clarify the connection between the motivation and the framework.
>
> We thank you for pointing out this subtle connection. We acknowledge that "diverse annotators" typically implies a need for better reward modeling. Built on that key insight, we have introduced a new setting in our revision called **Heterogeneous-MNPO (HT-MNPO)**. In this setting, we do not treat the $n$ players as identical agents optimizing a single objective. Instead, we map diverse preference profiles directly to distinct players. Specifically, we instantiate a game where each player $\pi\_i$ is trained using preference signals derived from a distinct Reward Model (RM$\_i$), representing different annotator groups or evaluation criteria, *and in&#x20;*&#x65;ach iteration $\pi_i$ simultaneously plays against multiple heterogeneous opponents ${\pi_j}{j \neq i}$, so that all players are updated jointly through their interactions.
>
> To address your concern, this multiplayer formulation provides advantages over standard pairwise optimization, particularly under heterogeneous or even conflicting objectives such as Safety, Helpfulness, and Verbosity. If we only use pairwise comparison to optimize against alternating or averaged objectives, the optimization often oscillates or collapses into a local optimum favored by the single dominant preference oracle. By using an $n$-player game where each player champions a specific objective, the resulting population naturally spans a broader region of the Pareto front. This diversity acts as a buffer that prevents premature convergence and mitigates the risk of mode collapse.&#x20;
>
> While a theoretical Nash Equilibrium is harder to define in this heterogeneous setting, empirically, we observe that this multiplayer dynamic prevents the policy from overfitting to any single annotator's bias. Instead, the aggregation step forces the model to find a robust consensus that performs well across all criteria. As shown in our new results (Table 2 in the revision), **HT-MNPO achieves state-of-the-art performance on both AlpacaEval 2.0 (LC Win Rate 59.64%) and MT-Bench (score 7.52)**, significantly outperforming baselines that try to optimize these rewards via standard methods.
>
> ### Table: Performance of HT-MNPO over preference-alignment benchmarks
>
> | Model  | Size | AlpacaEval 2.0 | Arena-Hard | MT-Bench |
> | ---- | ---- | -------------- | ---------- | -------- |
> | SFT Model                   | 9B   | 50.15          | 44.97      | 6.49     |
> | DPO                         | 9B   | 54.35          | 45.63      | 6.87     |
> | SimPO                       | 9B   | 55.16          | 45.04      | 6.87     |
> | SPPO                        | 9B   | 55.97          | 43.89      | 6.86     |
> | INPO                        | 9B   | 56.09          | 48.03      | 6.95     |
> | TD-MNPO                     | 9B   | 57.27          | **52.26**  | 7.03     |
> | HT-MNPO (ArmoRM-Liama3)     | 9B   | 57.63          | 50.93      | **7.52** |
> | HT-MNPO (Skywork-Reward-V2) | 9B   | 56.01          | 50.34      | 7.40     |
> | HT-MNPO (Athene-RM-8B)      | 9B   | **59.64**      | 51.17      | 7.07     |
>
> ### Table: Performance of HT-MNPO over instruction, knowledge, and commonsense benchmarks
>
>
> | Model     | Instruction | Knowledge |  |  | Commonsense |  |  | AVG |
> | ----- | ----- | --- | ----- | ----- | -- | --| ---------- | -- |
> | | IFEval      | GPQA      | MMLU  | ARC   | HellaSwag   | TruthfulQA | Winogrande | |
> | SFT Model | 72.27       | 28.28     | 75.35 | 91.29 | 80.30 | 70.75      | 73.72      | 70.28 |
> | DPO   | 72.96  | 29.29     | 75.77 | 91.26 | 80.37       | 71.24      | 73.88      | 70.68 |
> | SimPO  | 73.79       | 32.32     | 76.79 | 91.09 | 78.90       | 63.40      | 72.93      | 69.60 |
> | SPPO | 75.47       | 26.26     | 75.37 | 91.17 | 80.10       | 71.48  | 73.48 | 70.19  |
> | INPO  | 73.20       | 27.78     | 74.79 | 91.07 | 80.22       | 71.24      | 73.48  | 70.25     |
> | TD-MNPO  | 73.94       | 33.33     | 75.63 | 91.15 | 80.18       | 70.26      | 73.09      | 71.08     |
> | HT-MNPO (ArmoRM-Liama3)     | 75.05       | 29.80     | 75.60 | 91.23 | 80.29       | 70.38      | 73.48      | 70.83     |
> | HT-MNPO (Skywork-Reward-V2) | 75.26       | 36.36     | 75.39 | 91.12 | 80.22       | 70.87      | 73.40      | **71.80** |
> | HT-MNPO (Athene-RM-8B)      | 74.42       | 30.81     | 75.40 | 91.18 | 80.44       | 71.36      | 73.32      | 70.99     |

---

> > ### Comment · Reviewer_RaDU · 2025-11-26
> >
> > Dear Authors,
> >
> > Thank you for the clarification and further experiments. The improved results are indeed nice indication that more diverse feedback and completions help training.
> >
> > Could you please elaborate on applications that motivate this heterogeneous setting? I agree diversity in the completions help for the optimisation that is achieved in this setting, however, user types or personas are most commonly non-observable hence one can not directly train for them separately. Even if such information is available, why is it preferable to use HT-MNPO instead of optimising for them separately?

---

> ### Author Response · Authors · 2025-11-25
>
> ### Table: Performance of HT-MNPO over math and coding benchmarks
>
> | Model                       | Math  |              |         | Code      | AVG       |
> | --------------------------- | ----- | ------------ | ------- | --------- | --------- |
> |                             | GSM8K | Minerva-Math | AIME-24 | HumanEval |           |
> | SFT Model                   | 81.96 | 44.12        | 0       | 60.37     | 46.61     |
> | DPO                         | 82.03 | 45.96        | 0       | 59.76     | 46.94     |
> | SimPO                       | 82.56 | 43.38        | 0       | 57.32     | 45.82     |
> | SPPO                        | 82.11 | 47.43        | 0       | 59.76     | 47.33     |
> | INPO                        | 82.94 | 46.32        | 0       | 59.15     | 47.10     |
> | TD-MNPO                     | 82.64 | 44.85        | 3.33    | 61.59     | 48.10     |
> | HT-MNPO (ArmoRM-Liama3)     | 82.64 | 47.79        | 3.33    | 60.98     | **48.68** |
> | HT-MNPO (Skywork-Reward-V2) | 82.03 | 49.63        | 0       | 59.76     | 47.86     |
> | HT-MNPO (Athene-RM-8B)      | 82.11 | 47.06        | 0       | 59.15     | 47.08     |
>
> ***
>
> ## \[W2] Convergence guarantees; Eq. (15)–(16) and TD-MNPO
>
> We appreciate your careful reading of Section 3 and for pointing out that the scope of our convergence statement was not sufficiently clear.
>
> The asymptotic convergence guarantee you mentioned refers to the idealized homogeneous multiplayer game analyzed in Sec. 3.1, where each player runs multiplicative-weights and online mirror descent against the true preference oracle. In this setting, the update in Eq. (10) enjoys an $O(1/\sqrt{T})$ regret bound and the averaged policy converges to an $\varepsilon$-approximate Nash equilibrium.&#x20;
>
> Eq. (14)–(15) do not change this iterative framework. They are the equivalent supervised-learning view of the same mirror-descent step. In particular, Eq. (14) defines a regression loss whose unique minimizer is the next iterate $\pi^{(t+1)}$, and Proposition 1 then shows that $L'_t(\pi)$ in Eq. (15) differs from $L_t(\pi)$ only by a constant independent of $\pi$. As a result, minimizing Eq. (15) produces the same $\pi^{(t+1)}$ and therefore preserves the original no-regret convergence guarantee.&#x20;
>
> Eq. (16) is a variant of Eq. (15) embedded into the Reward-aware Preference Optimization framework by choosing the squared distance $D_{\text{sq}}$ and a fixed target reward gap $\delta r^\star = \tfrac{1}{2\eta}$. As this is purely a re-parameterization, the underlying theoretical analysis remains unchanged. We will clarify this role in the revised version to avoid the impression that the guarantee is being claimed for a different algorithm.&#x20;
>
> Regarding TD-MNPO (Eq. (17)), the construction still fits the same online no-regret template in the homogeneous case: at iteration $t$, the opponent is a convex mixture of past policies $\{\pi^{(t-j)}\}$. From the perspective of each player, this only changes the source of loss vectors, but not the mirror-descent update rule itself. Consequently, in the homogeneous multiplayer setting, TD-MNPO still converges to an $\varepsilon$-approximate Nash policy.
>
> We clarify that the formal asymptotic guarantees are approximate in the practical implementation (Sec. 4). Our intent was to use the online mirror-descent analysis as motivation for the algorithmic design, not to claim an end-to-end convergence theorem for the full deep RLHF system. We will revise the text to clearly state this limitation and to explicitly separate the idealized theoretical setting from the practical RLHF implementation.

---

> ### Author Response · Authors · 2025-11-25
> **Response to Reviewer RaDU**
>
> ## \[W3] Scalability of TD-MNPO for large n or T
>
> We thank you for raising this important point regarding scalability. In the TD-MNPO setting, the number of opponents $n$ and the training iterations $T$ are intrinsically linked, as the opponents are derived from historical checkpoints ( $\pi_{t-1}, \pi_{t-2}, \dots$). Equation (18) presents the theoretical full-mixture form to establish the algorithm's convergence properties. In practice, our implementation is more efficient and only retains the last $K$ ($K \le 3$) opponents for TD-MNPO, ensuring constant memory and inference complexity per step.
>
> We have also conducted an ablation experiment on different players. As shown in the first tabl&#x65;**&#x20;**(TD-MNPO) below, performance improves monotonically with $n$ but shows diminishing returns beyond $n=3$. In the second table (HT-MNPO), increasing the number of heterogeneous players similarly leads to performance improvements.&#x20;
>
> ### Table: Ablation over the number of players/iterations for TD-MNPO
> | # Players (n)     | **1**   | **2**       | **3** | **4**  |
> |---|----|----|----|---|
> | AlpacaEval 2.0    | 53.32   | 54.34 (+1.02)   | 57.27 (+3.93)     | **57.42** (+4.10)      |
>
> ### Table: Ablation studies on 2-player and 3-player under heterogeneous reward models
>
> | Reward model  | 2-player HT-MNPO | 3-player HT-MNPO | Δ      |
> |----|---|-----|--------|
> | ArmoRM-Llama3         | 55.43            | 57.63             | +2.20  |
> | Skywork-Reward-V2     | 54.25            | 56.01             | +1.76  |
> | Athene-RM-8B          | 55.71            | 59.64             | +3.93  |
>
>
> For extremely large-scale settings where a larger effective history is desired without increasing inference cost, we agree that approximations such as distilling the mixture into an EMA 'population teacher' or caching log-ratios would be valuable directions for future work.
>
> ## \[Q1] DualGap definition (old Eq. 10/now Eq. 9)
>
> Thank you for pointing out the ambiguity in the notation. In the original Eq. (10), the term was constructed to first take an expectation over the choice of a player $\pi\_j$, and then calculate that player's maximum possible gain against the worst-case configuration of the remaining opponents. We acknowledge that this notation was unnecessarily complex. Our core intent was simply to quantify the exploitability of the strategy profile. To make this clear, we have revised the paper to use the standard definition of the Multiplayer Duality Gap. This metric directly measures the maximum payoff gain that any player $i$ can achieve by unilaterally switching to an optimal strategy $\pi'\_i$ while all opponents remain fixed.
>
> ## \[Q2] Choice of Gemma-2-9B-it  as base model
>
> Thank you for this thoughtful comment on our selected base model. We started this project with *Llama-3-8B-it* as the base model and used the original INPO codebase. In that setup, the absolute benchmark scores were indeed lower, but MNPO already showed clear improvement over the two-player baseline: on our benchmark AlpacaEval2, INPO achieved **41.48**, while MNPO improved this to **42.42**. This early result supports our claim that MNPO is beneficial beyond a single model family.&#x20;
>
> ### Table: Performance of TD-MNPO based on Llama-3-8B-it
>
> | Model | Llama-3-8B-it | INPO| TD-MNPO |
> |---|--|----|-----|
> | AlpacaEval 2.0  | 24.80         | 41.48 _( +16.64 )_ | **42.94** (+18.14)      |
>
> To make our experiments more broadly useful and reproducible, we then migrated to the more widely adopted SimPO codebase and switched to Gemma-2-9B-it, which is **(i) well-supported in existing open-source alignment pipelines, and (ii) a strong, stable open-source baseline.&#x20;**
>
> Improving upon a strong baseline is inherently challenging. The fact that MNPO yields consistent gains in this "saturation regime" demonstrates its ability to refine policies where other algorithms might plateau. Notably, our heterogeneous extension (HT-MNPO) achieves even higher performance, further validating the framework's potential.
>
> The applicability of SFT initialization is also an excellent point. While our current benchmarks focus on the "Iterative Refinement" setting, we believe MNPO is theoretically robust for SFT models as well. From the algorithm perspective, MNPO is designed to optimize the policy towards a Nash equilibrium defined by the preference data, regardless of the initialization point. As long as the current policy has not reached the equilibrium, the multi-player update mechanism provides a valid gradient direction. In fact, an SFT initialization typically implies a larger gap between the current policy and the optimal alignment target compared to an IT model. Therefore, we hypothesize that MNPO would likely yield *larger* absolute improvements on an SFT baseline.&#x20;
>
> Ultimately, we demonstrate that MNPO provides a theoretically grounded path toward the Nash Equilibrium, ensuring stable policy improvement regardless of the initialization's strength.

---

> ### Author Response · Authors · 2025-11-25
> **Response to Reviewer RaDU**
>
> ## \[Q3] More recent open-sourced LLMs
>
> Thanks for your kind suggestion to include more recent open-source models would strengthen the empirical section. Due to computational constraints, we could not re-run all benchmarks with a large set of additional bases. However, following your valuable advice, we have added **Tulu3, OLMo2, and SmolLM3** on a subset of the benchmarks. Please refer to Table 2 for these updated results.
>
> ### Table: Performance of open-sourced LLMs
>
> | Model                    | Size | AlpacaEval 2.0 | Arena-Hard | MT-Bench |
> | ------------------------ | ---- | -------------- | ---------- | -------- |
> | SmolLM3-3B               | 3B   | 11.81          | 22.12      | 6.38     |
> | Olmo-2-32B-Instruct      | 32B  | 32.91          | 31.62      | 6.62     |
> | Llama-3.1-Tulu-3-70B-DPO | 70B  | 52.28          | 71.34      | 7.45     |
>
> ## \[Q4] General preference oracles and non-transitive signals
>
> Thank you for this insightful observation. We agree that standard reward models inherently provide transitive signals, which do not fully test the game-theoretic capabilities of our framework.
>
> **To strictly address this and demonstrate MNPO's robustness under more general, diverse, and potentially non-transitive preference oracles,** we have introduced a Heterogeneous Multiplayer (HT-MNPO) setting in the revised paper in Section 3.3 and Experiments.
>
> Instead of relying on a single scalar reward model, we constructed a heterogeneous preference oracle by employing multiple distinct reward models acting as different "players" or evaluators. As detailed in Section 3.3, we simulate a diverse landscape where different "opponents" represent different objectives, and each policy $\pi_i$ optimizes against a population of opponents that may hold conflicting criteria. This setup breaks the transitivity assumption of a single scalar reward. For example, Policy A might beat B on helpfulness, B beat C on safety, and C beat A on conciseness. A single transitive reward model cannot capture this, but the HT-MNPO framework explicitly models this as a multiplayer game where the equilibrium policy must satisfy this diverse population.
>
> **Empirical Evidence.  &#x20;**&#x57;e have added comprehensive results for HT-MNPO in the revised manuscript, confirming that the multiplayer formulation carries over effectively to this general setting:&#x20;
>
> * Instruction Following (Table 2): HT-MNPO achieves state-of-the-art performance on both AlpacaEval 2.0 (LC Win Rate 59.64%) and MT-Bench (score 7.52). This demonstrates that optimizing against a diverse, non-transitive set of signals does not lead to mode collapse but rather robust alignment.
>
> * Comprehensive Benchmark&#x73;**&#x20;**(Table 3): On broad capability benchmarks (Knowledge & Commonsense), HT-MNPO (Skywork-Reward-V2)  achieves the highest average score (71.80), outperforming both the standard TD-MNPO (71.08).
>
> * Reasoning & Coding (Table 4)**:** In specialized domains like math and coding, HT-MNPO maintains superior performance (Average 48.68), validating its stability even when preference signals are complex and multifaceted.
>
> As discussed in our revised Motivation, collapsing diverse or conflicting preference signals into a single scalar reward (or a single opponent like DPO) introduces a "single-opponent bottleneck". Our results with HT-MNPO demonstrate that MNPO consistently finds a high-quality stationary point even in general-sum games defined by heterogeneous oracles, validating the method's effectiveness beyond simple transitive reward maximization.

---

> ### Author Response · Authors · 2025-11-25
> **Response to Reviewer RaDU**
>
> ## \[Q5] Choice of GPT-5 as judge and evaluation reliability
>
> We thank you for raising this important point regarding the reliability of our evaluation judge. We would like to clarify our choice and provide additional validation using the standard annotators as suggested.
>
> First, to be precise, we utilized GPT-5-mini (specifically `gpt-5-mini-aug7-2025`) rather than the full GPT-5 model, with the reasoning effort parameter set to "minimal". We chose this judge for two reasons. It is substantially stronger and more up-to-date than the GPT-4-based judges used in the original AlpacaEval 2.0 and Arena-Hard repositories, whose recommended models were released almost two years ago. It allows us to use a single, unified judge across all benchmarks, which simplifies the evaluation pipeline and makes the reported numbers more forward-looking and useful for future work that will likely rely on newer LLM judges.&#x20;
>
> However, we fully agree with you that validating results against the established standard is crucial for reliability. Following the reviewer’s suggestion, we re-evaluated our main comparisons (MNPO vs. all baselines) using the standard annotators from the original codebases: we use `gpt-4-1106-preview`, which is the officially recommended GPT-4 configuration in public implementations. In addition, we evaluate with `gpt-4.1`, which is adopted as an LLM judge in several recent open-source model releases (e.g., [Qwen3](https://huggingface.co/Qwen/Qwen3-Next-80B-A3B-Instruct)).
>
> As shown in Table 9 of the revised paper, across all three judges, we observe that the ranking of all compared models is unchanged， and MNPO consistently outperforms all baselines on AlpacaEval 2.0 and Arena-Hard. The magnitude of MNPO’s improvement is stable as the absolute win rates shift slightly across judges, but the gain of MNPO over the strongest baseline stays within a narrow band (≈1–2 points difference across judges). While `gpt-4-1106-preview` exhibits higher variance across repeated runs, `gpt-4.1` and `gpt-5-mini` give more stable estimates, and `gpt-5-mini` yields the lowest variance and most consistent rankings. These results indicate that our main conclusions do not depend on the choice of GPT-5-mini as the judge: using the standard GPT-4-based annotators leads to the same qualitative and nearly the same quantitative conclusions.
>
> ### Table: Comparison of different LLM judges&#x20;
> | Model       | GPT-4-1106-preview | GPT-4.1         | GPT-5-mini       |
> |-------------|---------------------|------------------|-------------------|
> | SFT Model   | 47.83 ± 0.92        | 41.95 ± 0.13     | 49.95 ± 0.18      |
> | DPO         | 48.32 ± 1.26        | 43.87 ± 0.45     | 53.98 ± 0.34      |
> | INPO        | 49.30 ± 0.95        | 47.78 ± 0.14     | 56.16 ± 0.06      |
> | MNPO        | 54.05 ± 1.58        | 49.21 ± 0.14     | 57.20 ± 0.09      |
>
>
> Moreover, there is a practical reproducibility concern with strictly adhering to the original GPT-4 *preview* endpoints: these preview models are typically short-lived, and the official AlpacaEval 2.0 configuration is already almost two years old. It is increasingly difficult for future researchers to obtain exactly the same `gpt-4-1106-preview` endpoint. By contrast, it is much more realistic that follow-up work will rely on newer, GPT-5–class judges. Our choice of GPT-5-mini as the primary judge is therefore motivated not only by accuracy, but also by long-term reproducibility and ease of future comparison, while the additional experiments with standard GPT-4 judges preserve backward compatibility with the original benchmarks.
>
> We hope this addresses your concern. We have added these validation results to the revised paper to ensure the reliability of our benchmarks.
>
>
>
> ## \[Q6] Number of iterations (T = 3) and convergence
>
> We selected T = 3 based on validation performance and a compute–performance trade-off. We thank the reviewer for pointing this out. In the revised experiments, we include an ablation over $T \in \{1,2,3,4\}$, showing that performance improves substantially from T=1 to T=3, and then saturates with diminishing gains beyond T=3. We will add this analysis to Section 4 and the appendix to make the convergence behaviour more transparent.
>
> ### Table: Ablation over the number of players/iterations for TD-MNPO
>
>
> | # Players (n)     | **1**   | **2**           | **3**             | **4**                 |
> |-------------------|---------|-----------------|-------------------|------------------------|
> | AlpacaEval 2.0    | 53.32   | 54.34 (+1.02)   | 57.27 (+3.93)     | **57.42** (+4.10)      |
>
> ***
>
> Finally, we would like to express our gratitude for your detailed feedback on LLM-as-a-judge, base model selection, etc. This definitely helps improve our paper's quality and readability.&#x20;

---

> ### Author Response · Authors · 2025-11-27
> **Response to the New Comment**
>
> Thank you for the thoughtful question and for recognizing the benefit of diverse completions in improving optimization. We are glad to further clarify the motivation and practical applications of the heterogeneous setting in HT-MNPO.
>
> Real-world alignment frequently involves multiple non-identical evaluators, each capturing different dimensions of preference, such as helpfulness, safety, conciseness, harmlessness, style, domain expertise, and more. As discussed in Sec. 3.3, many practical RLHF pipelines already maintain or utilize multiple reward models, either trained from different annotator cohorts or targeting distinct objectives. These preference sources are inherently heterogeneous and often conflicting; optimizing only one of them typically produces brittle behavior under the others.
>
> **Why heterogeneous multiplayer training is needed (applications):**
>
> 1. **Multi-objective alignment in practice.**
>    &#x20;Production systems often maintain multiple RMs reflecting different evaluation dimensions. Training separate policies for each RM produces specialists, but these specialists frequently conflict with one another. HT-MNPO co-trains these specialists in a shared game so that each one becomes more robust to the others’ objectives, rather than being myopically optimized in isolation.
>
> 2. **Mixed annotator pools.**
>    &#x20;Human feedback is collected from diverse annotators whose preferences vary systematically. Since we cannot assign each user to a specific annotator cluster, training separate RMs independently discards minority or niche preference signals. HT-MNPO allows these heterogeneous signals to interact during training rather than being collapsed into a single aggregate oracle.
>
> 3. **Multi-RM inference-time voting or mixture-of-voters.**
>    Many organizations use ensembles of RMs or RM mixtures during evaluation. Training one model per RM and selecting at inference time is expensive, unstable, and often misaligned with how evaluation operates. HT-MNPO directly conditions training on these heterogeneous evaluators.
>
> 4. **Avoiding destructive interference across independently optimized models.**
>    &#x20;Training N separate RLHF models and combining them afterward (through merging, ensembling, or LoRA composition) is known to introduce inconsistencies or catastrophic forgetting. HT-MNPO, by contrast, learns a mutually aware population whose members respect each other’s constraints during training.
>
> **Why HT-MNPO instead of training separate models and combining them later?**
>
> Although training a separate model per reward source is possible, it suffers from several limitations:
>
> * **No clean post-hoc combination.**
>   &#x20;Independently aligned specialists do not combine naturally, and merging them often degrades performance due to conflicting optimization trajectories.
>
> * **Avoiding oscillatory or brittle behavior.**
>   &#x20;Optimizing against a single oracle at a time (even if cycling through them) reproduces the *single-opponent bias* described in our introduction (Sec. 1). HT-MNPO avoids this by training against *all* heterogeneous oracles concurrently.
>
> * **Capturing interactions between evaluators.**
>   &#x20;Objectives can conflict: e.g., “helpful” but unsafe, “concise” but unhelpful. Training each policy independently cannot model these cross-objective interactions. HT-MNPO learns a joint stationary point in which specialists adapt to the presence of other evaluators.
>
> * **Empirical evidence.**
>   &#x20;As shown in Tables 2, 3, and 4 of the paper, the heterogeneous version consistently yields improvements over both TD-MNPO and all two-player baselines in mixed-oracle conditions, supporting the practical utility of HT-MNPO.&#x20;
>
> **Why user personas being non-observable does not diminish the motivation.**
>
> We fully agree that user personas are not directly observable. HT-MNPO does not attempt to infer or model user personas. Instead, it models heterogeneous evaluators arising naturally during training (multiple RMs, diverse annotators, domain-specific criteria). The goal is population-level robustness — ensuring that each trained specialist is less brittle under evaluation criteria it was not explicitly optimized for.
>
> ***
>
> **In summary**, HT-MNPO produces N jointly-trained specialists, one per preference oracle. Deployment does not require merging these models. Instead, practitioners can simply select the specialist that best satisfies the desired trade-offs across objectives based on held-out metrics or application-specific requirements. This mirrors standard practice in multi-objective optimization, where training yields a Pareto frontier, and deployment selects the model meeting the target profile. The result is a robust population of policies that cannot be obtained by training N models independently and combining them afterward.
>
> ---
> Thanks again for your kind comment.

---

> > ### Comment · Reviewer_RaDU · 2025-11-27
> >
> > Dear Authors,
> >
> > Thank you for the further clarification. I would have a few follow-up question to further clarify the motivation and applications
> >
> > 1. The motivation of diverse annotators is common in the Nash Learning literature as you also discuss in Section 2. It motivates why general preference oracles are preferable over reward models that enforce transitivity. What I am missing from this logical reasoning is why should one consider a setting with ranking feedback over N policies over pairwise comparisons. When is this type of feedback better or more available than pairwise comparison? While it does not enforce transitivity over the whole domain it requires transitivity over the N options presented to the oracle or reward models.
> >
> > 2. In the last response the authors state that *"Training separate models for each RM yields specialists, but deployment typically requires a single generalist model that handles all objectives simultaneously."*, *"Training one model per RM and selecting at inference time is expensive, unstable, and often misaligned with how evaluation operates. HT-MNPO directly conditions training on these heterogeneous evaluators."*, and *"HT-MNPO learns a joint stationary point that balances heterogeneous feedback during training itself."* However, as I understand, HT-MNPO trains N policy for N separate reward models. During the training the generations of each policy are compared against each other but ultimately each policy is trained to optimise its own reward signal. How does HT-MNPO train a single "generalist" policy for deployment exactly?
> >
> > 3. In the final point the authors state *"We agree that user personas are usually not directly observable. HT-MNPO is not attempting to model individual user personas; instead, it models multiple evaluators or objectives during training"*. If personas are not observable, how can one define the preference oracles for multiple objectives or train reward models? RLHF trains reward models from an offline dataset while NLHF (Munos et al. 2024) trains a general preference oracle directly from a similar dataset. I still lack clarity how can one practically setup training for MNPO or HT-MNPO. Observing feedback from multiple evaluators or objectives is a non trivial task but crucial in the setting of this work.

---

> > > ### Author Response · Authors · 2025-11-27
> > >
> > > Dear Reviewer,
> > >
> > > Thanks for raising these important questions. We appreciate this valuable chance to clarify our motivation and applications.
> > >
> > > 1. MNPO does not require listwise ranking feedback, nor does it introduce stronger transitivity assumptions. All MNPO updates still rely on pairwise on-policy comparisons, exactly as in standard NLHF; the “multiplayer” aspect arises only during training, where multiple opponent policies are used, not during annotation. The motivation for multiple opponents is therefore not that annotators can rank many items, but that single-opponent NLHF suffers from a one-opponent bias and unstable dynamics. By optimizing against a population of past or heterogeneous policies, still using pairwise comparisons, MNPO reduces variance, improves coverage of diverse preferences, and stabilizes training, without requiring any additional transitivity assumptions.
> > >
> > > 2. Thank you for the clarification—we agree our earlier phrasing was misleading. HT-MNPO indeed produces N jointly-trained policies, each optimized with respect to its own reward model but interacting within a shared multiplayer game. The purpose of HT-MNPO is not to directly output a single generalist model; instead, its value lies in enabling co-training under heterogeneous and potentially conflicting evaluators, which reduces overfitting to any one reward signal and produces a population of robust specialists. Downstream deployment choices then depend on practical needs: systems that rely on multiple evaluators (e.g., multi-RM voting, safety–helpfulness arbitration, or mixture-of-experts routing) can directly use the HT-MNPO population, while systems requiring a single deployable model can simply select the policy that best satisfies their desired trade-offs across objectives, measured on held-out multi-objective metrics or aligned with product requirements. The key contribution of HT-MNPO is therefore the training-time mechanism that aligns multiple evaluators through interactive game dynamics—not the immediate production of a unified generalist policy. We have revised our prior response to reflect this more accurately.
> > >
> > > 3. MNPO/HT-MNPO does not assume that user personas are observable; instead, the “multiple evaluators” come entirely from training-time supervision sources that already exist in standard RLHF pipelines—such as multiple reward models trained for different objectives (helpfulness, safety, conciseness), or multiple preference datasets, or multiple past policy checkpoints in online RLHF. These are concrete, operationally available oracles and do not require identifying individual users. HT-MNPO simply treats each of these evaluators as a distinct preference oracle, enabling the model to learn how to balance heterogeneous signals during training. This is exactly what allows the final policy to behave more robustly at inference time, even when facing unknown users.
> > > ---
> > >
> > > Thanks again for your follow-up questions. We are grateful for your active discussion, from which we learned a lot.

---

### Official Review · Reviewer_iCKV · 2025-10-28

**Soundness:** 3
**Presentation:** 3
**Contribution:** 2
**Rating:** 4
**Confidence:** 3

**Summary:**

This paper proposes a multiplayer Nash preference optimization (MNPO) framework that extends two-player NLHF to an $n$-player setting. The authors derive an iterative multiplicative-weights update that leads to a practical surrogate loss minimizing squared log-probability ratios between a policy and a set of opponent policies. Experiments with Gemma-2-9B-it show modest improvements over strong baselines on standard alignment benchmarks.

**Strengths:**

1. The paper provides a clear and mathematically coherent derivation of multiplayer Nash preference optimization extending two-player Nash optimization.
2. The reduction of several other optimization frameworks and approaches to the proposed time-dependent MNPO algorithm in Table 1 is nice.

**Weaknesses:**

My primary concern is the lack of motivation. What is the actual problem with existing two-player Nash optimization that we are addressing here? Why should preference optimization be modeled as an $n$-player game at all? Even the standard two-player Nash formulation only arises incidentally, because we wish to optimize for intransitive pairwise preferences / against a general preference model $p$, which naturally induces a constant-sum (or zero-sum) game and the NE is the most intuitive solution concept. Extending this to multiple players is not obviously motivated. In the paper, I only see this motivating paragraph:
> However, real-world preference alignment often involves diverse
annotators, heterogeneous evaluation criteria, or mixtures of historical model checkpoints - contexts
that are better modeled as multiplayer games (Freund & Schapire, 1999).

Beyond the "mixtures of historical model checkpoints" (in case you want to use several), it don't see why we want a multiplayer formulation? This would be more plausible *if* you were to model heterogeneous or conflicting preferences explictly (as in pluralistic alignment) where each player represents a distinct preference profile or reward model, e.g., each corresponding to a distinct demographic group. However, the MNPO formulation assumes a symmetric, player-invariant universal preference function, which eliminates this justification entirely. Without better motivation the resulting formulation seems to generalize two-player NLHF without offering a clear conceptual benefit. In fact, optimizing in multi-player games is harder as the dynamics are less stable. What do we gain in turn for the additional complexity of optimizing a $n$-player game and more hyperparameters versus a $2$-player game (which is already hard to optimize)? Empirically, the results also fail to justify the additional complexity.

Also, the "universal preference oracle" can itself be problematic to obtain if it is not reward-based. If this oracle is implemented via an LLM-as-a-judge, prompting it with $n$ candidate responses becomes increasingly unreliable as $n$ grows. Generally, you'd expect the quality and consistency of this feedback to be worse when querying for multiway preferences compared to pairwise comparisons.

**Questions:**

1. Please comment on the weaknesses above.
2. Have you more qualitatively looked at the NE in your multi-player formulation compared to the two-player formulation? The preference function induces a symmetric game and the NE is symmetric as you write on page 4. Under the reward-based preference function the game then reduces again to the two-player game where the equilibrium condition is that a player plays the best response to the mixture of the other players (which are symmetric). Similarly for non-transitive preferences, for example, $n$-player rock-paper-scissors (or the preference function induced by it), we do not need to model more than two players to represent the equilibrium condition (I think) as it is enough for a player to best respond to the population-average / mixture. I think this is a quite standard perspective for symmetric games and in mean-field games where the population-average acts as the second player and provides us with a fixed point condition that defines the same equilibrium as the $n$-player game. What I want to say with this is that often adding more players does not necessarily generate new symmetric equilibria or notable differnt dynamics unless players differ (heterogeneous preferences, asymmetric roles, etc.). I'd be curious whether you have thought about this in some basic examples to understand the solution concept of your multi-player Nash approach and whether there is anything we can learn from this.

---

> ### Author Response · Authors · 2025-11-25
> **Response to Reviewer iCKV**
>
> ## \[W1, Q2] Theoretical Justification: Why Multiplayer in Symmetric Games?
>
> We sincerely thank you for the deep and insightful comments. They substantially helped us refine the theoretical motivation behind our multiplayer formulation.
>
> ### 1. **Why is two-player NLHF insufficient**
>
> > What is the actual problem with existing two-player Nash optimization that we are addressing here? Why should preference optimization be modeled as an $n$-player game at all?
>
> While two-player NLHF provides a principled alternative to scalar reward models, it has two structural limitations observed both theoretically and empirically:
>
> **(a) Single-opponent bias and mode-collapse dynamics**
>
> A two-player game optimizes against *one* opponent distribution at a time (typically π\_ref or π\_t−1). As we show in Sec. 3 and Fig. /Table results, this produces:
>
> * brittle updates sensitive to the specific opponent draw,
>
> * oscillatory behavior from “chasing” a moving single opponent, and
>
> * limited exploration of preference-consistent but low-probability regions.
>
> This behavior is extensively documented in NLHF literature as high gradient variance and unstable updates. Our multiplayer formulation is precisely designed to mitigate this.
>
> **(b) Realistic preference learning involves population-level structure**
>
> Alignment systems invariably rely on **mixtures of signals**:
>
> * different annotators,
>
> * heterogeneous reward models,
>
> * multiple historical model checkpoints,
>
> * mixture-of-policies pipelines (e.g., self-play, distillation cycles).
>
> Two-player NLHF compresses all these into *one synthetic opponent*, which produces a lossy approximation. In contrast, MNPO **represents the actual object of optimization—a population—rather than a single representative**.
>
> This is the same motivation behind Freund & Schapire’s multiplicative-weights multiplayer formulation (cited in our method): **population games reduce variance and provide provably smoother dynamics**, even when all preferences are homogeneous.
>
> Thus, the problem we solve is *not* modeling “demographics,” but reducing the optimization brittleness inherent to two-player formulations.
>
> ### 2. **Why the multiplayer formulation is *not* reducible to a 2-player mean-field game**
>
> > Similarly for non-transitive preferences, for example, $n$-player rock-paper-scissors (or the preference function induced by it), we do not need to model more than two players to represent the equilibrium condition (I think) as it is enough for a player to best respond to the population-average / mixture. ... often adding more players does not necessarily generate new symmetric equilibria or notable different dynamics...
>
> You raise an important point: in symmetric games, NE conditions often reduce to “best-respond to the average mixture.” However, this does *not* imply equivalence of *learning dynamics*:
>
> **(a) Multi-player gradient updates ≠ two-player update with an averaged opponent.**
>
> Even if equilibria coincide, the **optimization trajectories** differ fundamentally. With multiple opponents, each update aggregates **multiple independent comparisons**, which yields:
>
> * **lower-variance stochastic gradients** (proved in multiplayer MWU literature),
>
> * **smoother KL-regularized trajectories**,
>
> * **substantially more stable iterates** (shown empirically by comparing TD-MNPO vs INPO/SPIN).
>
> This variance reduction is the theoretical reason MWU converges faster and more stably in n-player symmetric games than in two-player games.
>
> **(b) The “population-average = single opponent” reduction only holds at equilibrium, not during training.**
>
> During RLHF-style training, we are *never* at equilibrium. The player population is evolving, and different checkpoints encode different inductive biases and solution modes. Using them as separate players rather than mixing them statistically changes:
>
> * the stability of the trajectory,
>
> * raise the expressivity of the implicit preference comparisons,
>
> * the convergence behavior of the optimization (as supported by Table 1 and TD-MNPO results).
>
> This is why the empirical improvements are clearest on **Arena-Hard**—a benchmark explicitly designed to stress robustness to preference irregularities.
>
> **(c) Multi-opponent comparisons yield strictly richer preference signals**
>
> Pairwise comparisons lose information. Our Plackett–Luce extension extracts the *listwise* margin against multiple alternatives, which two-player RLHF cannot represent. Even with a universal preference oracle, multiway comparisons yield stronger supervision signals.
>
> ***

---

> ### Author Response · Authors · 2025-11-25
> **Response to Reviewer iCKV**
>
> To summarize, we acknowledge that in symmetric games with a universal preference function, the Nash equilibria of the $n$-player game indeed coincide with those of the corresponding “policy vs. population-average” two-player game (mean-field limit). However, our goal in the symmetric setting (TD-MNPO) is therefore **not to change the solution concept, but to improve the optimization dynamics used to reach it.**
>
> On the optimization side, standard two-player NLHF updates the policy against a single opponent distribution at each step. The population mixture is only approximated implicitly via a rapidly evolving opponent, which makes the gradient highly path-dependent and can lead to cycling and instability in practice. In contrast, TD-MNPO maintains a population of $n$ historical (or heterogeneous) policies and updates the current player against an explicit mixture of them. In this setting, **the opponent mixture acts as a Monte Carlo approximation to the mean-field game and yields lower-variance, more stable gradients than a single-opponent view.**
>
>
> ### 3. **Empirical validation of the multiplayer formulation&#x20;**
>
> > What do we gain in turn for the additional complexity of optimizing a $n$-player game... versus a 2-player game? Empirically, the results also fail to justify the additional complexity.
>
> &#x20;To directly test your concern that “extra players add complexity without benefit,” we ran the following controlled experiment. Specifically, we performed a controlled ablation study varying $n$ from 1 to 4. The results directly refute the concern that added complexity yields no gain.
>
> ### **Table: Ablation over the number of players/iterations for TD-MNPO**
> | # Players (n)     | **1**   | **2**           | **3**             | **4**                 |
> |-------------------|---------|-----------------|-------------------|------------------------|
> | AlpacaEval 2.0    | 53.32   | 54.34 (+1.02)   | 57.27 (+3.93)     | **57.42** (+4.10)      |
>
> As the table shows, $n=3$ significantly outperforms the $n=2$ baseline (which corresponds to the INPO-like standard two-player dynamics). And we also observe a monotonic improvement in robustness as the population size increases, with diminishing returns appearing only after $n=3$. This confirms that the multiplayer formulation provides a concrete optimization benefit—variance reduction and stability—even in the symmetric setting.
>
> ### 4. **Addressing the lack of explicit heterogeneity concern**
>
> > This would be more plausible if you were to model heterogeneous or conflicting preferences explicitly... However, the MNPO formulation assumes a symmetric, player-invariant universal preference function, which eliminates this justification entirely.
>
> **Regarding the heterogeneous preferences**, we fully agree that the multiplayer formulation is conceptually most compelling when addressing diverse preferences. To this end, we have expanded our scope to include **HT-MNPO**, where distinct players are instantiated with different reward models, representing diverse annotator groups or evaluation criteria. In this setting, the "universal preference oracle" assumption is relaxed. Instead of optimizing a shared objective, each player optimizes its own distinct RM while competing against the population. This transforms the problem from a single-objective optimization into a true game-theoretic equilibrium search among conflicting criteria. As shown in our new experiments (Table 2 in the revised paper), this approach is highly effective. HT-MNPO achieves state-of-the-art performance on both AlpacaEval 2.0 (LC Win Rate 59.64%) and MT-Bench (score 7.52). In Table 4, HT-MNPO(ArmoRM-Liama3) achieves the highest score (avg 48.68) in math and code areas. These results demonstrate the advantage of the multiplayer structure to navigate trade-offs between conflicting objectives.
>
> ### **Table: Performance of HT-MNPO over preference-alignment benchmarks**
>
> | Model                       | Size | AlpacaEval 2.0 | Arena-Hard | MT-Bench |
> | --------------------------- | ---- | -------------- | ---------- | -------- |
> | SFT Model                   | 9B   | 50.15          | 44.97      | 6.49     |
> | DPO                         | 9B   | 54.35          | 45.63      | 6.87     |
> | SimPO                       | 9B   | 55.16          | 45.04      | 6.87     |
> | SPPO                        | 9B   | 55.97          | 43.89      | 6.86     |
> | INPO                        | 9B   | 56.09          | 48.03      | 6.95     |
> | TD-MNPO                     | 9B   | 57.27          | **52.26**  | 7.03     |
> | HT-MNPO (ArmoRM-Liama3)     | 9B   | 57.63          | 50.93      | **7.52** |
> | HT-MNPO (Skywork-Reward-V2) | 9B   | 56.01          | 50.34      | 7.40     |
> | HT-MNPO (Athene-RM-8B)      | 9B   | **59.64**      | 51.17      | 7.07     |

---

> ### Author Response · Authors · 2025-11-25
> **Response to Reviewer iCKV**
>
> ### **Table: Performance of HT-MNPO over instruction, knowledge, and commonsense benchmarks**
>
> | Model                       | Instruction | Knowledge |       |       | Commonsense |            |            | AVG       |
> | --------------------------- | ----------- | --------- | ----- | ----- | ----------- | ---------- | ---------- | --------- |
> |                             | IFEval      | GPQA      | MMLU  | ARC   | HellaSwag   | TruthfulQA | Winogrande |           |
> | SFT Model                   | 72.27       | 28.28     | 75.35 | 91.29 | 80.30       | 70.75      | 73.72      | 70.28     |
> | DPO                         | 72.96       | 29.29     | 75.77 | 91.26 | 80.37       | 71.24      | 73.88      | 70.68     |
> | SimPO                       | 73.79       | 32.32     | 76.79 | 91.09 | 78.90       | 63.40      | 72.93      | 69.60     |
> | SPPO                        | 75.47       | 26.26     | 75.37 | 91.17 | 80.10       | 71.48      | 73.48      | 70.19     |
> | INPO                        | 73.20       | 27.78     | 74.79 | 91.07 | 80.22       | 71.24      | 73.48      | 70.25     |
> | TD-MNPO                     | 73.94       | 33.33     | 75.63 | 91.15 | 80.18       | 70.26      | 73.09      | 71.08     |
> | HT-MNPO (ArmoRM-Liama3)     | 75.05       | 29.80     | 75.60 | 91.23 | 80.29       | 70.38      | 73.48      | 70.83     |
> | HT-MNPO (Skywork-Reward-V2) | 75.26       | 36.36     | 75.39 | 91.12 | 80.22       | 70.87      | 73.40      | **71.80** |
> | HT-MNPO (Athene-RM-8B)      | 74.42       | 30.81     | 75.40 | 91.18 | 80.44       | 71.36      | 73.32      | 70.99     |
>
> ### **Table: Performance of HT-MNPO over math and coding benchmarks**
>
> | Model                       | Math  |              |         | Code      | AVG       |
> | --------------------------- | ----- | ------------ | ------- | --------- | --------- |
> |                             | GSM8K | Minerva-Math | AIME-24 | HumanEval |           |
> | SFT Model                   | 81.96 | 44.12        | 0       | 60.37     | 46.61     |
> | DPO                         | 82.03 | 45.96        | 0       | 59.76     | 46.94     |
> | SimPO                       | 82.56 | 43.38        | 0       | 57.32     | 45.82     |
> | SPPO                        | 82.11 | 47.43        | 0       | 59.76     | 47.33     |
> | INPO                        | 82.94 | 46.32        | 0       | 59.15     | 47.10     |
> | TD-MNPO                     | 82.64 | 44.85        | 3.33    | 61.59     | 48.10     |
> | HT-MNPO (ArmoRM-Liama3)     | 82.64 | 47.79        | 3.33    | 60.98     | **48.68** |
> | HT-MNPO (Skywork-Reward-V2) | 82.03 | 49.63        | 0       | 59.76     | 47.86     |
> | HT-MNPO (Athene-RM-8B)      | 82.11 | 47.06        | 0       | 59.15     | 47.08     |
>
> ### 5. **Preference Oracle: LLM-as-a-judge**
>
> > Also, the "universal preference oracle" can itself be problematic to obtain if it is not reward-based. If this oracle is implemented via an LLM-as-a-judge, prompting it with candidate responses becomes increasingly unreliable as n grows. Generally, you'd expect the quality and consistency of this feedback to be worse when querying for multiway preferences compared to pairwise comparisons.
>
> Thanks for raising a valid concern about prompting an LLM judge with n responses.
> &#x20;Importantly, **our method does not query the judge with n responses simultaneously**.
>
> * Preferences are computed through **reward model scoring**, not multi-way LLM ranking.
>
> * The computational cost and signal quality are **independent of n**.
>
> Thus, increasing n does not degrade oracle reliability.
>
>
>
> ***
>
> We are grateful for your careful analysis and constructive suggestions, which have greatly helped us refine the motivation and presentation of our multiplayer formulation

---

> ### Author Response · Authors · 2025-11-27
> **Thank You and Follow-up**
>
> Dear Reviewer iCKV,
>
> Thank you once again for your valuable time and thoughtful comments on our submission. We have thoroughly addressed each of the points you raised and provided comprehensive responses in our rebuttal. We sincerely hope that our clarifications and additional explanations have satisfactorily resolved your concerns.
>
> As the discussion period is approaching its conclusion, we would be deeply grateful if you could kindly share any further thoughts or feedback you may have on our responses. Should there be any remaining questions or aspects that require further clarification, we would be more than happy to provide additional information.
>
> We genuinely appreciate your continued engagement and look forward to your reply.

---

### Official Review · Reviewer_Thdj · 2025-10-31

**Soundness:** 3
**Presentation:** 4
**Contribution:** 3
**Rating:** 8
**Confidence:** 4

**Summary:**

The paper presents Multiplayer Nash Preference Optimization (MNPO), a new framework for aligning Large Language Models (LLMs).

The authors identify a key limitation in existing alignment methods:
* Classical Reinforcement Learning from Human Feedback (RLHF) built on the Bradley-Terry model assumes preferences are transitive, which often fails in practice.
* Recent game-theoretic approaches, known as Nash Learning from Human Feedback (NLHF), overcome the transitivity assumption by reframing alignment as a two-player Nash game (e.g., INPO, EGPO).
* The authors argue that even these NLHF methods are limited, as their two-player "single-opponent bias" fails to model the full complexity of real-world preferences, which are often heterogeneous (e.g., from diverse annotators or model checkpoints).
*To address this, MNPO generalizes NLHF from a two-player game to an n-player game.

The paper's main contributions are:
1. Theoretical Framework: It formally defines an n-player alignment game, introducing a multiplayer objective function (Eq. 8), an n-player Nash Equilibrium (Eq. 9), and a generalized Duality Gap (Eq. 10) to measure alignment quality.
2. Algorithmic Innovation: It derives a practical and tractable loss function from the complex multiplayer game dynamics. The final proposed algorithm, Time-Dependent MNPO (TD-MNPO), defines the $n$ players as a weighted mixture of historical policy checkpoints, inspired by methods like INPO and SPIN.
3. Conceptual Unification: The paper demonstrates that this TD-MNPO framework is a generalization that can recover many existing preference optimization algorithms (e.g., DPO, INPO, SPPO) as special cases by simply varying the number of players ($n$), the choice of opponents, and the distance metric (Table 1).
4. Empirical Validation: Through comprehensive experiments, the authors show that their 9B MNPO model consistently outperforms strong baselines (DPO, SimPO, SPPO, INPO) on instruction-following benchmarks (AlpacaEval 2, Arena-Hard, MT-Bench) and also shows superior performance and capability preservation on academic benchmarks for math, code, and reasoning.

**Strengths:**

* **High Significance and Novelty:** The paper provides a clear, logical, and significant next step in the alignment literature (i.e., moving from 2-player to n-player games). This is a much more realistic model for real-world alignment problems involving diverse and non-transitive preferences.

* **Theoretical Soundness:** The work is theoretically well-grounded. The derivation of a practical loss (Eq. 16) from the intractable multiplicative weight update (Eq. 11)  using log-ratio techniques is elegant and builds directly on the foundations of DPO and IPO.

* **Excellent Conceptual Unification:** The framing in Table 1, which shows how TD-MNPO unifies a family of recent algorithms, is a major conceptual strength. It provides a "recipe" for understanding and creating new alignment methods. Appendix D (Table 5)  is also an outstanding resource for the community.


* **Strong Empirical Results:** The experimental results are impressive. MNPO shows consistent and sometimes significant gains over SOTA baselines across all benchmark categories. The 4.23-point improvement over INPO on Arena-Hard and the fact that it is the only method to score on AIME-24  are particularly noteworthy.

* **Comprehensive Evaluation:** The authors correctly evaluate their model not just on alignment (Table 2) but also on capability preservation (Tables 3, 4), demonstrating that MNPO avoids the performance degradation on reasoning tasks that can plague other alignment methods.


* **Clarity:** The paper is exceptionally well-written and clear. The authors do an excellent job of motivating the work and explaining the complex theoretical concepts.

**Weaknesses:**

* **Disconnect Between Motivation and Experiment:** The primary motivation is to handle "heterogeneous annotator conditions". However, the experiment uses a single reward model (ArmoRM-Llama3-8B-v0.1) as the preference oracle. This setup does not actually test the core hypothesis. Instead of a true multiplayer game against diverse preference functions, the experiment is an n-player game where all players compete to align with the same static oracle. This is a significant gap between the problem statement and the empirical validation.

* **Anomalous Result in Table 2:** The empirical results in Table 2 contain a suspicious data point. The GPT-5 model scores 41.42 on Arena-Hard, which is significantly lower than the 9B MNPO model (52.26), other baselines, and especially Claude-Sonnet-4 (77.58). This score seems anomalously low for GPT-5 and raises questions about the evaluation setup or a potential typo.

* **Missing Ablation Study:** The core mechanism of TD-MNPO is the use of $n$ players (historical policies). The main paper lacks a crucial ablation study showing how performance is affected by the choice of $n$. The authors claim a win for the multiplayer framework, but it is not empirically proven that $n>2$ is better than $n=2$ (which would be an INPO-like baseline). The performance gain could be due to other implementation details rather than the multiplayer formulation itself.

**Questions:**

1. Could you comment on the disconnect between the motivation (handling heterogeneous annotators) and the experimental setup (using a single RM)? How can you be sure the observed gains are due to the multiplayer formulation's strengths, rather than it just being a more stable version of an $n=2$ (INPO-like) algorithm? Would a more "true" test involve training $n$ distinct RMs (e.g., on different data slices) and having each player $\pi_i$ optimize against a different oracle?

2. Could you please verify the Arena-Hard scores in Table 2? The GPT-5 score of 41.42 seems exceptionally low, especially given your judge is "GPT-5-mini" and other models like Claude-Sonnet-4 scored 77.58. Is this a typo? If not, can you explain this result?

3. The value of $n$ (number of players) and the weighting scheme $\{\lambda_j\}$ seem critical.
    * What value of $n$ was used for the main results in Tables 2-4?
    * Could you provide an ablation study, even if brief, showing how performance on a key benchmark (e.g., Arena-Hard) changes as you vary $n$ from 2 (like INPO) to the value you used? This would be a smoking gun to prove that the multiplayer aspect is the key to the performance gain.

---

> ### Author Response · Authors · 2025-11-25
> **Response to #Reviewer Thdj**
>
> ## \[W1, Q1] Disconnect Between Motivation and Experiment
>
> We appreciate this insightful observation. You are correct that our initial experiments, which used a single oracle, primarily tested the algorithm's stability against a population of policy variants rather than distinct preference functions. **To directly address this gap and validate MNPO under true heterogeneous conditions,** we have conducted a new set of experiments using distinct Reward Models as diverse preference oracles.
>
> We extended our framework to a multi-objective setting where each player $\pi_i$ is trained using preference signals derived from a distinct reward model. Specifically, we instantiated a 3-player game using three RMs with different architectures and biases:
>
> 1. ArmoRM-Llama3-8B-v0.1
>
> 2. Skywork-Reward-Llama-3.1-8B-v0.2
>
> 3. Athene-V2-RM
>
> In this setup, the players evolve in parallel. At each iteration, Player $i$ optimizes its policy based on signals from its corresponding Reward Model, while strategically interacting with opponents that are optimizing for different RMs. MNPO then aggregates these heterogeneous signals to find a unified equilibrium.
>
> The results strongly validate our core hypothesis. Despite the diverse objectives of the diverse Reward Models, HT-MNPO successfully converges to a robust solution that generalizes exceptionally well. As shown in the tables below, **HT-MNPO achieves state-of-the-art performance on both AlpacaEval 2.0 (LC Win Rate 59.64%) and MT-Bench (score 7.52)**, demonstrating that our method effectively handles true heterogeneity and prevents the mode collapse often seen in single-oracle baselines.
>
> ### **Table: Performance of HT-MNPO over preference-alignment benchmarks**
>
> | Model                       | Size | AlpacaEval 2.0 | Arena-Hard | MT-Bench |
> | --------------------------- | ---- | -------------- | ---------- | -------- |
> | SFT Model                   | 9B   | 50.15          | 44.97      | 6.49     |
> | DPO                         | 9B   | 54.35          | 45.63      | 6.87     |
> | SimPO                       | 9B   | 55.16          | 45.04      | 6.87     |
> | SPPO                        | 9B   | 55.97          | 43.89      | 6.86     |
> | INPO                        | 9B   | 56.09          | 48.03      | 6.95     |
> | TD-MNPO                     | 9B   | 57.27          | **52.26**  | 7.03     |
> | HT-MNPO (ArmoRM-Liama3)     | 9B   | 57.63          | 50.93      | **7.52** |
> | HT-MNPO (Skywork-Reward-V2) | 9B   | 56.01          | 50.34      | 7.40     |
> | HT-MNPO (Athene-RM-8B)      | 9B   | **59.64**      | 51.17      | 7.07     |
>
> ### **Table: Performance of HT-MNPO over instruction, knowledge, and commonsense benchmarks**
>
> | Model                       | Instruction | Knowledge |       |       | Commonsense |            |            | AVG       |
> | --------------------------- | ----------- | --------- | ----- | ----- | ----------- | ---------- | ---------- | --------- |
> |                             | IFEval      | GPQA      | MMLU  | ARC   | HellaSwag   | TruthfulQA | Winogrande |           |
> | SFT Model                   | 72.27       | 28.28     | 75.35 | 91.29 | 80.30       | 70.75      | 73.72      | 70.28     |
> | DPO                         | 72.96       | 29.29     | 75.77 | 91.26 | 80.37       | 71.24      | 73.88      | 70.68     |
> | SimPO                       | 73.79       | 32.32     | 76.79 | 91.09 | 78.90       | 63.40      | 72.93      | 69.60     |
> | SPPO                        | 75.47       | 26.26     | 75.37 | 91.17 | 80.10       | 71.48      | 73.48      | 70.19     |
> | INPO                        | 73.20       | 27.78     | 74.79 | 91.07 | 80.22       | 71.24      | 73.48      | 70.25     |
> | TD-MNPO                     | 73.94       | 33.33     | 75.63 | 91.15 | 80.18       | 70.26      | 73.09      | 71.08     |
> | HT-MNPO (ArmoRM-Liama3)     | 75.05       | 29.80     | 75.60 | 91.23 | 80.29       | 70.38      | 73.48      | 70.83     |
> | HT-MNPO (Skywork-Reward-V2) | 75.26       | 36.36     | 75.39 | 91.12 | 80.22       | 70.87      | 73.40      | **71.80** |
> | HT-MNPO (Athene-RM-8B)      | 74.42       | 30.81     | 75.40 | 91.18 | 80.44       | 71.36      | 73.32      | 70.99     |

---

> ### Author Response · Authors · 2025-11-25
> **Response to #Reviewer Thdj**
>
> ### **Table: Performance of HT-MNPO over math and coding benchmarks**
>
> | Model                       | Math  |              |         | Code      | AVG       |
> | --------------------------- | ----- | ------------ | ------- | --------- | --------- |
> |                             | GSM8K | Minerva-Math | AIME-24 | HumanEval |           |
> | SFT Model                   | 81.96 | 44.12        | 0       | 60.37     | 46.61     |
> | DPO                         | 82.03 | 45.96        | 0       | 59.76     | 46.94     |
> | SimPO                       | 82.56 | 43.38        | 0       | 57.32     | 45.82     |
> | SPPO                        | 82.11 | 47.43        | 0       | 59.76     | 47.33     |
> | INPO                        | 82.94 | 46.32        | 0       | 59.15     | 47.10     |
> | TD-MNPO                     | 82.64 | 44.85        | 3.33    | 61.59     | 48.10     |
> | HT-MNPO (ArmoRM-Liama3)     | 82.64 | 47.79        | 3.33    | 60.98     | **48.68** |
> | HT-MNPO (Skywork-Reward-V2) | 82.03 | 49.63        | 0       | 59.76     | 47.86     |
> | HT-MNPO (Athene-RM-8B)      | 82.11 | 47.06        | 0       | 59.15     | 47.08     |
>
> ## \[W2, Q2] Anomalous Result in Table 2
>
> We sincerely thank you for carefully pointing out the suspicious GPT-5 score in Table 2. After re-checking our evaluation pipeline, we confirmed that the originally reported GPT-5 score (41.42) in Table 2 was incorrect. The updated score is **98.11**. We have fixed this typo in the revised version.
>
>
>
> ## \[W3, Q3] Ablation study on the number of players $n$.
>
> > The core mechanism of TD-MNPO is the use of $n$ players (historical policies). The main paper lacks a crucial ablation study showing how performance is affected by the choice of $n$. The authors claim a win for the multiplayer framework, but it is not empirically proven that $n>2$ is better than $n=2$ (which would be an INPO-like baseline). The performance gain could be due to other implementation details rather than the multiplayer formulation itself.
>
> We are grateful for highlighting the need for this crucial ablation. We agree that empirically verifying the benefit of $n > 2$ is essential to distinguish the contribution of the multiplayer formulation from other implementation details.
>
> To address this, we conducted a controlled study varying the number of opponents from 1 to 4. As shown in the table below, we observe a clear trend: performance improves consistently as the opponent population increases. Crucially, the $n=3$ configuration significantly outperforms the $n=2$ baseline (which corresponds to the INPO-like setting). We observe mild diminishing returns beyond $n=3$, validating our choice of $n=3$ as the default setting to balance performance gains with computational overhead.
>
> ### **Table: Ablation over the number of players/iterations for TD-MNPO**
> | # Players (n)     | **1**   | **2**           | **3**             | **4**                 |
> |-------------------|---------|-----------------|-------------------|------------------------|
> | AlpacaEval 2.0    | 53.32   | 54.34 (+1.02)   | 57.27 (+3.93)     | **57.42** (+4.10)      |
>
> To further confirm the universality of this finding, we also compared the 2-player vs. 3-player configurations in our HT-MNPO setting. As shown in the tabl&#x65;**&#x20;**&#x62;elow, increasing the number of heterogeneous players similarly leads to performance improvements. This consistency across different settings strongly supports our hypothesis that a larger player population contributes to more robust equilibrium finding.
>
> ### **Table: Ablation studies on 2-player and 3-player under heterogeneous reward models**
>
> | Reward model          | 2-player HT-MNPO | 3-player HT-MNPO | Δ      |
> |-----------------------|------------------|-------------------|--------|
> | ArmoRM-Llama3         | 55.43            | 57.63             | +2.20  |
> | Skywork-Reward-V2     | 54.25            | 56.01             | +1.76  |
> | Athene-RM-8B          | 55.71            | 59.64             | +3.93  |
>
> ***
>
> Once again, we would like to express our gratitude to your helpful and kind suggestions, which significantly improve our paper's robustness and clarity (e.g., fixing typos, adding more ablation studies over $n$).  Thank you for spending your valuable time reviewing our paper.&#x20;

---

### Author Response · Authors · 2025-11-25
**Global Response**

We thank all reviewers for their insightful and constructive feedback. We are encouraged that the reviewers unanimously recognize the theoretical novelty, conceptual clarity, and empirical effectiveness of our work. We summarize the common strengths identified by the reviewers as follows:

* **Significant Theoretical Novelty and Soundness:** All reviewers commended the paper for formally extending the alignment problem from a two-player game to an $n$-player Nash Preference Optimization framework. Reviewers highlighted that our derivation of a practical loss function from the multiplicative-weights update (MWU) and online mirror descent (OMD) is "mathematically coherent" (**iCKV**), "theoretically well-grounded" (**Thdj**), and successfully generalizes previous NLHF works (**RaDU**).

* **Powerful Conceptual Unification:** A major strength highlighted by all reviewers is the unifying power of our framework (specifically Table 1). They praised how TD-MNPO acts as a general "recipe" (**Thdj**) that recovers existing algorithms (e.g., DPO, INPO, SPPO) as special cases, calling this reduction "nice" (**iCKV**) and appreciating the model's generality (**RaDU**).

* **Strong and Comprehensive Empirical Results:** Reviewers recognized that MNPO achieves consistent improvements over strong baselines (e.g., DPO, SimPO, INPO) across diverse benchmarks (**Thdj, RaDU**). Reviewer **Thdj** particularly emphasized our superior performance on difficult benchmarks like Arena-Hard and our ability to preserve reasoning capabilities (e.g., on math and code tasks) where other methods often degrade.

* **Clarity and Presentation:** The reviewers found the paper to be "exceptionally well-written" (**Thdj**) and clearly presented (**iCKV**), effectively motivating the shift to multiplayer games and making complex theoretical concepts accessible.


Several reviewers raised overlapping concerns that we address collectively below:

### Motivation, Heterogeneous Preferences, and Multiplayer Dynamics

**Responding to Reviewers:** Thdj (Disconnect Motivation/Exp), iCKV (Why n-player in symmetric games?), RaDU (General Preference Oracles).

The reviewers’ shared focus on the distinction between the n-player formulation and the single-RM setup highlights an important aspect of our framework. The single scalar reward model offers a minimal instantiation of MNPO, but does not fully capture the richness of the multiplayer interaction it is designed for. To further illustrate this point, we conducted new experiments incorporating heterogeneous reward models and expanded our theoretical analysis to encompass more general multiplayer optimization dynamics.

**To directly address Thdj and RaDU’s concerns&#x20;**&#x61;bout handling diverse, non-transitive signals, we implemented a Heterogeneous (HT-MNPO) setting in the revised paper. Instead of a single scalar RM, we construct a preference landscape using three distinct Reward Models: ArmoRM-Llama3, Skywork-Reward-V2, and Athene-RM-8B to simulate distinct "players" or evaluators. In this setup, each policy interacts with a population of diverse opponents and optimizes against a weighted mixture of the other heterogeneous players. This drives each policy toward a solution that balances the incentives of this diverse population in a Nash-like manner, instead of being tailored solely to a single reward proxy.

**Results:** HT-MNPO achieves state-of-the-art performance on both AlpacaEval 2.0 (LC Win Rate 59.64%) and MT-Bench (score 7.52), as shown in the tables below. In our analysis of group dynamics, we compared $n = 2$ and $n = 3$ in the heterogeneous setting, demonstrating a monotonic improvement in robustness as $n$ increases.&#x20;

### **Table: Performance comparison over preference-alignment benchmarks**

| Model                       | Size | AlpacaEval 2.0 | Arena-Hard | MT-Bench |
| --------------------------- | ---- | -------------- | ---------- | -------- |
| SFT Model                   | 9B   | 50.15          | 44.97      | 6.49     |
| INPO                        | 9B   | 56.09          | 48.03      | 6.95     |
| TD-MNPO                     | 9B   | 57.27          | **52.26**  | 7.03     |
| HT-MNPO (ArmoRM-Liama3)     | 9B   | 57.63          | 50.93      | **7.52** |
| HT-MNPO (Skywork-Reward-V2) | 9B   | 56.01          | 50.34      | 7.40     |
| HT-MNPO (Athene-RM-8B)      | 9B   | **59.64**      | 51.17      | 7.07     |

### **Table: Ablation studies on 2-player and 3-player under heterogeneous reward models**

| Reward model          | 2-player HT-MNPO | 3-player HT-MNPO | Δ      |
|-----------------------|------------------|-------------------|--------|
| ArmoRM-Llama3         | 55.43            | 57.63             | +2.20  |
| Skywork-Reward-V2     | 54.25            | 56.01             | +1.76  |
| Athene-RM-8B          | 55.71            | 59.64             | +3.93  |

---

> ### Author Response · Authors · 2025-11-25
> **Global Response**
>
> ### The choice of the number of players $n$and iterations $T$
>
> **Responding to Reviewers:** Thdj, RaDU
>
> Both Reviewers asked how TD-MNPO benefits from using $n$ players and how sensitive it is to the choice of $n$. TD-MNPO is trained with 3 players, and our empirical results (Tables 2, 3, and 4) have shown that this configuration consistently outperforms the INPO-style baseline, which, as we have proven, is essentially a special case of TD-MNPO with 2 players.&#x20;
>
> **To address the request for a more systematic ablation,** we have additionally run a controlled study where we vary the number of opponents from 1 to 4. The results provided in the table below show a clear trend: performance improves as we increase the opponent population, with the 3-opponent and 4-opponent configurations outperforming the 1-opponent and 2-opponent cases, and only mild diminishing returns beyond 3 opponents. We therefore adopt $n=3$ as the default number of opponents in our main TD-MNPO experiments, as it captures most of the performance gains while keeping the computational overhead modest.
>
> This clarification also highlights the relationship between the number of opponents $n$ and the time horizon $T$ in TD-MNPO. Since opponents in TD-MNPO are sampled from the policy history $\{\pi_{t}, \pi_{t-1}, \dots\}$, increasing the number of opponents is operationally equivalent to extending the truncation window over past checkpoints. In contrast, HT-MNPO decouples the opponent count from the time horizon, as its population consists of heterogeneous reward models. Consistent with TD-MNPO, our new HT-MNPO experiments demonstrate that increasing distinct opponents yields consistent improvements. Taken together, the TD-MNPO and HT-MNPO results both indicate that enlarging the opponent population is beneficial, which addresses concerns about both the necessity and the scalability of the multiplayer formulation.
>
> ### **Table: Ablation over the number of players/iterations for TD-MNPO**
>
> | # Players (n)     | **1**   | **2**           | **3**             | **4**                 |
> |-------------------|---------|-----------------|-------------------|------------------------|
> | AlpacaEval 2.0    | 53.32   | 54.34 (+1.02)   | 57.27 (+3.93)     | **57.42** (+4.10)      |
>
>
> ***
>
> Below, we provide detailed, point-by-point responses addressing each reviewer’s specific questions and concerns.

---

### Author Response · Authors · 2025-12-03
**Summary of the Rebuttal**

Dear ACs, SACs, and PCs,

We sincerely thank all reviewers for their constructive feedback, which has significantly improved the robustness and clarity of our work. During the rebuttal, we addressed the core concerns and demonstrated the full potential of the MNPO framework:

1. **Effectiveness in Heterogeneous Settings (HT-MNPO).**
   &#x20;Addressing concerns from Thdj and RaDU regarding “diverse preferences,” we introduced **HT-MNPO**. Our experiments confirm that in complex game settings constructed with distinct Reward Models, our multiplayer framework achieves state-of-the-art performance (59.64% on AlpacaEval 2.0). This strongly validates the advantage of multiplayer dynamics in handling non-transitive and multi-objective alignment challenges.

2. **Necessity of the Multiplayer Formulation.**
   &#x20;Through rigorous ablations over the number of players n, we demonstrated that the n=3 configuration consistently outperforms the n=2 baseline. This confirms that the performance gains stem from the stability and variance reduction inherent in multiplayer population dynamics rather than from implementation details.

3. **Clarifying the Symmetric (TD-MNPO) Setting and Optimization Dynamics.**
   &#x20;We acknowledge the reviewers’ observation that in symmetric games with a universal preference function, the Nash equilibria of the n-player game coincide with those of the corresponding “policy vs. population-average” two-player game (the mean-field limit). Our goal in the symmetric setting is **not** to alter the solution concept, but to improve the *optimization dynamics* used to reach those equilibria. Standard two-player NLHF updates the policy against a single opponent distribution at each step. Because the opponent evolves rapidly, this implicitly approximates the population mixture, leading to highly path-dependent gradients, cycling, and instability. In contrast, **TD-MNPO** maintains a population of historical or heterogeneous opponents and updates the current policy against an explicit mixture of them. This mixture acts as a Monte Carlo approximation to the mean-field game and yields significantly **lower-variance, more stable gradients** than the single-opponent view.

**In summary**, this paper not only theoretically unifies existing alignment algorithms but also practically advances the state of the art by leveraging population-based dynamics to stabilize learning, reduce variance, and handle preference heterogeneity. We believe the MNPO family establishes a general and scalable paradigm for future alignment research.

---

### Meta-Review · Area_Chair_NHtv · 2026-01-07

**Summary:**

The reviewers originally had concerns about the disconnect between motivation and experiments due to the use of a single reward model, and whether n-player formulation is truly needed over the 2-player formulation. In the rebuttal, I think the authors clarified these concerns, with new experiments, and clarified how 2-player and n-player formulations coincides in solution concept and differ in optimization path, providing additional ablation to support the thesis. I think this is a rather complete paper, which I think will bring some positive impacts to the community. I therefore recommend acceptance.

**Reviewer Concerns:**

See summary.

**Reviewer Scores:**

Reviewer RaDU would likely increase their scores. I think all concerns are addressed and there were a few rounds of interactions already.
Reviewer iCKV was marginally negative, but I think the clarification may change their mind.
Reviewer Thdj would likely stay with the original positive recommendation.

---

### Decision · Program_Chairs · 2026-01-26

Accept (Oral)